# SP140–RESIST pathway regulates interferon mRNA stability and antiviral immunity

Kristen C. Witt[1,2,3], Adam Dziulko[4], Joohyun An[2,3], Filip Pekovic[5], Arthur Xiuyuan Cheng[2,3], Grace Y. Liu[1,2,3], Ophelia Vosshall Lee[2,3], David J. Turner[5], Azra Lari[6], Moritz M. Gaidt[2,3,8], Roberto Chavez[1,2,3], Stefan A. Fattinger[2,3], Preethy Abraham[2,3], Harmandeep Dhaliwal[7], Angus Y. Lee[7], Dmitri I. Kotov[2,3], Laurent Coscoy[2,3], Britt A. Glaunsinger[1,3,6], Eugene Valkov[5], Edward B. Chuong[4] & Russell E. Vance[1,2,3,7 ✉]

Type I interferons are essential for antiviral immunity[1] but must be tightly regulated[2]. The conserved transcriptional repressor SP140 inhibits interferon-β (*Ifnb1*) expression through an unknown mechanism[3,4]. Here we report that SP140 does not directly repress *Ifnb1* transcription. Instead, SP140 negatively regulates *Ifnb1* mRNA stability by directly repressing the expression of a previously uncharacterized regulator that we call RESIST (regulated stimulator of interferon via stabilization of transcript; previously annotated as annexin 2 receptor). RESIST promotes *Ifnb1* mRNA stability by counteracting *Ifnb1* mRNA destabilization mediated by the tristetraprolin (TTP) family of RNA-binding proteins and the CCR4–NOT deadenylase complex. SP140 localizes within punctate structures called nuclear bodies that have important roles in silencing DNA-virus gene expression in the nucleus[3]. Consistent with this observation, we find that SP140 inhibits replication of the gammaherpesvirus MHV68. The antiviral activity of SP140 is independent of its ability to regulate *Ifnb1*. Our results establish dual antiviral and interferon regulatory functions for SP140. We propose that SP140 and RESIST participate in antiviral effector-triggered immunity[5,6].

Type I interferons (IFN-I) are cytokines that have central roles in antiviral immunity[1], autoimmunity[2] and cancer[7]. IFN-I include *Ifnb1* and numerous *Ifna* and other isoforms, which signal through the IFNα receptor (IFNAR) to induce hundreds of interferon-stimulated genes that counter infection[8]. The pathways leading to *Ifnb1* induction have been enumerated in detail[1]. However, despite a longstanding appreciation that *Ifnb1* mRNAs turn over rapidly in cells[9], relatively little is known about the pathways controlling *Ifnb1* mRNA stability. This is surprising given that mRNA turnover is a critical point of regulation for many other cytokines[10,11]. Moreover, substantial evidence points to the importance of negative regulation of IFN-I, as excessive IFN-I can drive autoimmunity[2] and susceptibility to bacterial infections[1].

SP140 is an evolutionarily conserved but poorly characterized member of the speckled protein (SP) family of epigenetic readers[3] that contain histone-recognition and/or DNA-binding domains, as well as an oligomerization domain structurally homologous to caspase-activation and recruitment domains. SP family members including SP140 form nuclear bodies (NBs)—punctate structures that orchestrate various nuclear functions, including transcriptional regulation[3,12]. The precise functions of SP140 are unclear, although previous research suggests that SP140 is a transcriptional repressor that is essential for macrophage and, possibly, T cell function[3,13–20]. In humans, loss-of-function mutations in *SP140* are associated with immune disorders such as multiple sclerosis and B cell cancers[21–26]. We previously found a critical role for

SP140 in the repression of IFN-I in vivo, as *Sp140*[−/−] mice are highly susceptible to multiple bacterial infections due to elevated IFN-I[4]. However, the mechanism by which SP140 represses IFN-I is entirely unknown.

## SP140 inhibits *Ifnb1* mRNA stability

To investigate how SP140 negatively regulates IFN-I, we first characterized *Ifnb1* transcript levels in wild-type (WT) C57BL/6J (B6) and *Sp140*[−/−] bone-marrow-derived macrophages (BMMs) stimulated with agonists of distinct IFN-I-inducing pathways, including bacterial lipopolysaccharide (LPS), the dsRNA mimic poly(I:C) and the mouse STING agonist DMXAA (Fig. 1a). We noted that *Ifnb1* transcript levels were similar between B6 and *Sp140*[−/−] cells at early timepoints (2–4 h) after stimulation, but remained elevated (around tenfold) in *Sp140*[−/−] cells at late timepoints (8 h) for all IFN-I inducing stimuli (Fig. 1a). A time-course analysis of *Ifnb1* transcript levels after DMXAA stimulation confirmed that *Sp140*[−/−] cells displayed increased *Ifnb1* mRNA at late timepoints after stimulation (8–12 h) (Fig. 1b).

We hypothesized that the elevated levels of *Ifnb1* mRNA at late timepoints in *Sp140*[−/−] BMMs are due to the increased stability of *Ifnb1* transcripts. To assess the stability of *Ifnb1* mRNA, we used Roadblock quantitative PCR with reverse transcription (RT–qPCR)[27], a method in which the nucleotide analogue 4-thiouridine (4SU) is added to cells and incorporated into newly transcribed mRNAs. Isolated RNA is then

[1]Howard Hughes Medical Institute, University of California, Berkeley, CA, USA. [2]Division of Immunology and Molecular Medicine, University of California, Berkeley, CA, USA. [3]Department of Molecular and Cell Biology, University of California, Berkeley, CA, USA. [4]Department of Molecular, Cellular, and Developmental Biology and BioFrontiers Institute, University of Colorado Boulder, Boulder, CO, USA. [5]National Cancer Institute, National Institutes of Health, Frederick, MD, USA. [6]Department of Plant & Microbial Biology, University of California, Berkeley, CA, USA. [7]Cancer Research Laboratory, University of California, Berkeley, CA, USA. [8]Present address: Research Institute of Molecular Pathology, Vienna BioCenter, Vienna, Austria. ✉e-mail: rvance@berkeley.edu

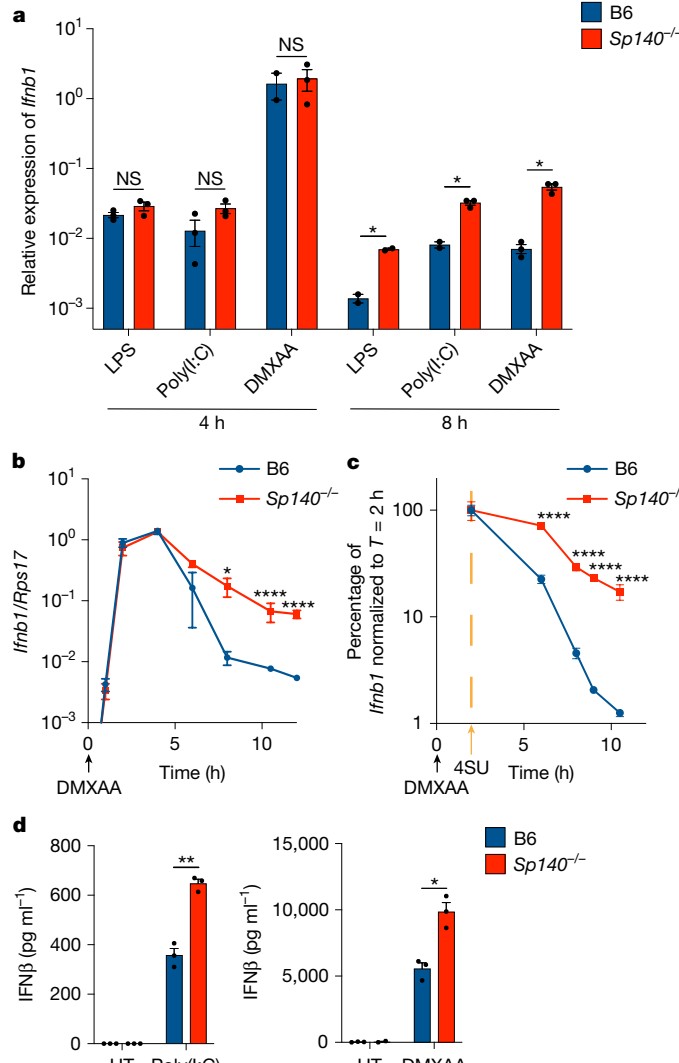

**Fig. 1 | *Ifnb1* mRNA is stabilized in the absence of SP140. a**, RT–qPCR analysis of *Ifnb1* in BMMs treated for 4 or 8 h with 10 ng ml$^{-1}$ LPS, or 100 µg ml$^{-1}$ poly(I:C) or DMXAA. $P = 0.26$ (LPS), $P = 0.17$ (poly(I:C)) and $P = 0.79$ (DMXAA) at $T = 4$ h; and $P = 0.02$ (DMXAA) at $T = 8$ h. **b**, RT–qPCR analysis of *Ifnb1* from BMMs treated with 100 µg ml$^{-1}$ DMXAA at the indicated timepoints. $P > 0.9999$ ($T = 0$ and 4 h), $P = 0.986$ ($T = 1$ h), $P = 0.995$ ($T = 2$ h), $P = 0.0062$ ($T = 6$ h) and $P < 0.0001$ ($T = 8$, 10.5 and 12 h). **c**, Roadblock RT–qPCR analysis of BMMs treated with 4SU 2 h after treatment with 100 µg ml$^{-1}$ DMXAA. $P = 0.999$ ($T = 2$ h) and $P < 0.0001$ (all other timepoints). **d**, Enzyme-linked immunosorbent assay (ELISA) analysis of IFNβ protein in the supernatants of BMMs treated for 24 h with 100 µg ml$^{-1}$ poly(I:C) or DMXAA. $P = 0.00185$ (poly(I:C)) and $P = 0.020$ (DMXAA). Representative experiments are shown from two independent experiments. $n = 3$ (**b**–**d** and **a** (DMXAA 8 h, LPS 4 h, poly(I:C) 4 h, *Sp140*$^{-/-}$ + DMXAA 4 h, *Sp140*$^{-/-}$ + poly(I:C) 8 h)) and $n = 2$ (**a** (B6 + DMXAA 4 h, LPS 8 h, B6 + poly(I:C) 8 h)) wells. Data are mean ± s.e.m. Statistical analysis was performed using two-tailed *t*-tests with Welch's correction (**d**) and false-discovery rate (FDR) correction (**a**), or two-way analysis of variance (ANOVA) with Šidák's multiple-comparison correction (**b** and **c**); *$P < 0.05$, **$P < 0.005$, ***$P < 0.0005$, ****$P < 0.0001$; NS, not significant. The statistical test results are provided in the Source data.

treated with *N*-ethylmaleimide (NEM), which reacts with 4SU to introduce a sterically bulky group that blocks reverse transcription. By adding 4SU at the peak of *Ifnb1* induction ($t = 2$ h after DMXAA stimulation), we could follow the subsequent decay of existing *Ifnb1* mRNA without detecting newly transcribed mRNAs. Using this approach, we found that the decay of *Ifnb1* transcript was markedly delayed in *Sp140*$^{-/-}$ BMMs (Fig. 1c). The increased stability of *Ifnb1* transcript in *Sp140*$^{-/-}$ BMMs

resulted in an increase in IFNβ protein levels in the culture supernatants (Fig. 1d). These results demonstrate an unexpected role for SP140 in the regulation of *Ifnb1* transcript stability.

## SP140 represses RESIST

SP140 lacks predicted RNA-binding domains and is believed to act in the nucleus to repress transcription[3]. We therefore hypothesized that SP140 indirectly regulates *Ifnb1* mRNA stability by repressing the transcription of an unknown factor. To identify this factor, we generated RNA sequencing (RNA-seq) data from DMXAA-treated B6 and *Sp140*$^{-/-}$ BMMs, and from DMXAA-treated *Ifnar*$^{-/-}$ and *Sp140*$^{-/-}$*Ifnar*$^{-/-}$ BMMs. The latter dataset eliminates the potentially confounding effects of elevated IFN-I signalling in *Sp140*$^{-/-}$ BMMs. Notably, few genes were differentially expressed between DMXAA-treated *Ifnar*$^{-/-}$ and *Sp140*$^{-/-}$*Ifnar*$^{-/-}$ BMMs (Fig. 2a). Other than *Ifnb1* itself, no known IFN-I regulators were differentially expressed, and Gene Ontology analysis of differentially expressed genes (DEGs) did not identify significantly enriched biological processes. Notably, only two DEGs correlated with *Ifnb1* upregulation across both RNA-seq datasets (Fig. 2b): *Sp140*, which was downregulated in *Sp140*$^{-/-}$ cells as expected, and *Gm21188*, a poorly annotated gene that was upregulated in *Sp140*$^{-/-}$ cells. *Gm36079*, a tandem paralogue of *Gm21188*, was upregulated in both datasets, but only significantly so in IFNAR-sufficient *Sp140*$^{-/-}$ BMMs that were treated with DMXAA. The open reading frames of *Gm21188* and *Gm36079* differ by a single silent nucleotide substitution and therefore encode an identical 20.9 kDa protein. On the basis of our results below, we refer to this protein as RESIST (regulated stimulator of interferon via stabilization of transcript), and refer to the *Gm21188* and *Gm36079* genes as *Resist1* and *Resist2* (*Resist1/2*), respectively.

To test whether *Resist1/2* are direct targets of SP140, we performed CUT&RUN analysis using an anti-HA antibody in DMXAA-treated *Sp140*$^{-/-}$ BMMs transduced with HA–SP140, or with untagged SP140 as a negative control. We confirmed that untagged and HA-tagged SP140 were functional and able to reduce late *Ifnb1* transcript levels induced by DMXAA (Extended Data Fig. 1a). In parallel, to characterize how SP140 regulates chromatin accessibility at target genes, we also generated assay for transposase-accessible chromatin with sequencing (ATAC–seq)[28] data from DMXAA-treated B6 and *Sp140*$^{-/-}$ BMMs. Consistent with previous SP140 chromatin immunoprecipitation followed by sequencing (ChIP–seq) results[15], we found that SP140 binds to genetic loci involved in development, such as *Hoxa9* (Extended Data Fig. 1b–d). Similarly, we confirmed that SP140 generally represses chromatin accessibility at target genes, consistent with previous studies implicating SP140 in transcriptional repression[14,15,17] (Extended Data Fig. 1c). SP140 binding also correlated with the transcriptionally repressive histone mark H3K27me3 in publicly available ChIP–seq datasets (Extended Data Fig. 1e). However, a gene set previously reported to drive macrophage dysfunction after *SP140* knockdown[14,15] was not significantly upregulated in *Sp140*$^{-/-}$ BMMs (Fig. 2a and Extended Data Fig. 1f), suggesting that these genes do not explain the elevated IFN-I induction that occurs in the absence of SP140. Consistent with our hypothesis that SP140 regulates *Ifnb1* mRNA stability and does not directly regulate *Ifnb1* transcription, SP140 did not bind to or regulate chromatin accessibility at the *Ifnb1* gene (Extended Data Fig. 2a) or known *Ifnb1* regulatory elements[29–32] (Extended Data Fig. 2b).

Importantly, SP140 robustly bound and repressed chromatin accessibility at the *Resist1/2* locus (Fig. 2b,c). Of the significantly upregulated genes in DMXAA-treated *Sp140*$^{-/-}$*Ifnar*$^{-/-}$ versus *Ifnar*$^{-/-}$ BMMs, only *Resist1* was both bound by SP140 and showed increased chromatin accessibility in the absence of SP140 (Fig. 2b,c). SP140 binds to an approximately 10 kb region encompassing *Resist1/2*, and negatively regulates chromatin accessibility at *Resist1/2* gene bodies (Fig. 2c). These results demonstrate that *Resist1/2* are directly repressed by

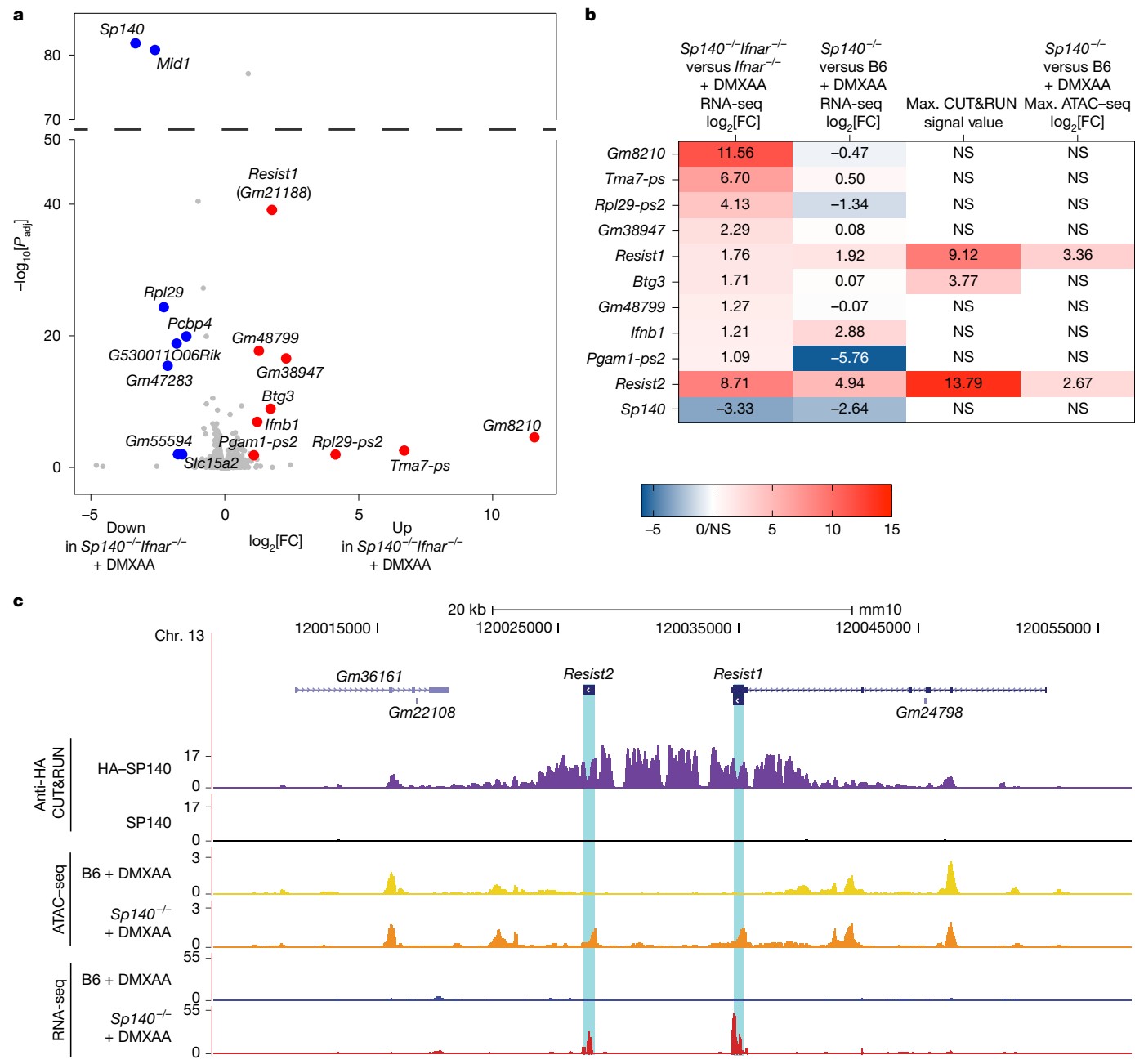

**Fig. 2 | *Resist1* and *Resist2* are repressed by SP140 and correlate with increased *Ifnb1* transcripts in *Sp140*$^{-/-}$ cells. a**, DEGs from RNA-seq data of DMXAA-treated *Sp140*$^{-/-}$ *Ifnar*$^{-/-}$ versus *Ifnar*$^{-/-}$ BMMs. Genes indicated in red are upregulated in *Sp140*$^{-/-}$ *Ifnar*$^{-/-}$ BMMs with log$_2$[fold change (FC)] > 1 and adjusted $P$ ($P_{adj}$) < 0.05. Genes indicated in blue are downregulated in *Sp140*$^{-/-}$ *Ifnar*$^{-/-}$ BMMs with log$_2$[FC] > −1 and $P_{adj}$ < 0.05. The $P_{adj}$ value for *Sp140* is <2.225 × 10$^{-308}$ and is graphed as −10 × $P_{adj}$ of *Mid1* for visualization. *Resist2* (*Gm36079*) is not depicted on the volcano plot as it is removed by the DeSeq2 independent filtering function for genes with low read counts. $P_{adj}$ values are provided in the Source data and were calculated using two-tailed Wald tests

with Benjamini−Hochberg correction for multiple comparisons using the DeSeq2 package. **b**, The maximum HA−SP140 CUT&RUN MACS2 signal values, the maximum log$_2$[FC] in chromatin accessibility from ATAC-seq of DMXAA-treated B6 and *Sp140*$^{-/-}$ BMMs, and the log$_2$[FC] from RNA-seq analysis of DMXAA-treated B6, *Sp140*$^{-/-}$, *Ifnar*$^{-/-}$ and *Sp140*$^{-/-}$ *Ifnar*$^{-/-}$ BMMs for significantly upregulated DEGs from **a**, as well as *Sp140* and *Resist2*. Cells are coloured according to the column value. **c**, Alignment of reads at the *Resist1/2* locus from anti-HA CUT&RUN data for DMXAA-treated BMMs transduced with HA−SP140 or SP140, and ATAC−seq/RNA-seq data of DMXAA-treated B6 and *Sp140*$^{-/-}$ BMMs. Alignments were visualized in the UCSC genome browser.

SP140, and are the only direct SP140 target genes detectably upregulated along with *Ifnb1* in *Sp140*$^{-/-}$ macrophages.

## RESIST enhances *Ifnb1* mRNA stability

The RESIST protein has not been characterized in mice, but it is the orthologue of the human annexin 2 receptor (ANXA2R), which is encoded by a single gene (*ANXA2R*). In WT mice, *Resist1* is detectably

expressed in myeloid cells such as neutrophils, macrophages, monocytes and eosinophils (Extended Data Fig. 3a). In human peripheral blood mononuclear cells, *ANXA2R* is detectably expressed in multiple immune cell populations, including monocytes, natural killer cells, dendritic cells, innate lymphoid cells, and T and B cells[33] (Extended Data Fig. 3b). ANXA2R was first proposed as a receptor for annexin 2 (ANXA2) based on the observation that ANXA2R overexpression increased the binding of exogenous ANXA2 to the cell surface[34]. However, ANXA2R is

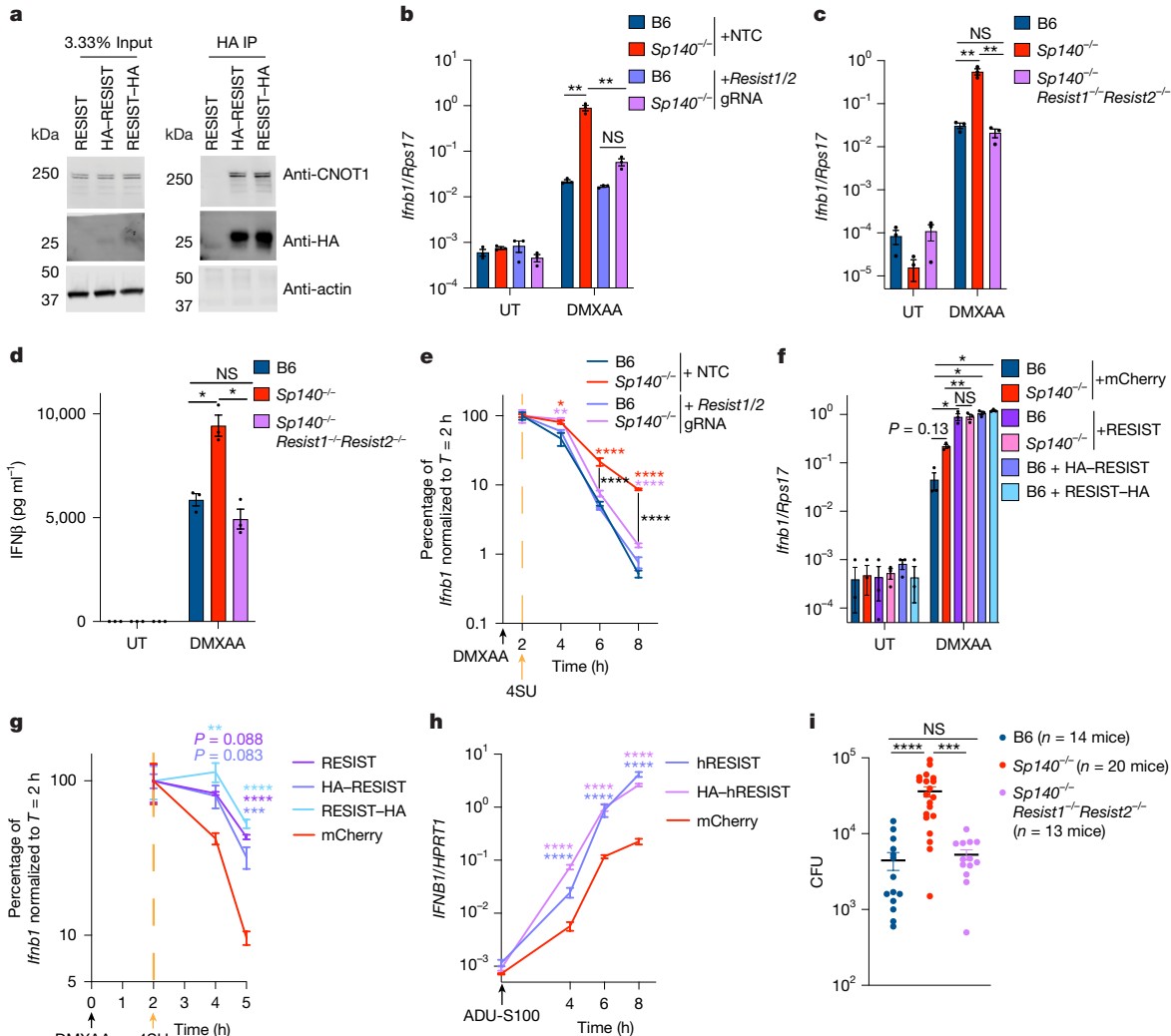

**Fig. 3 | RESIST binds to the CCR4–NOT complex and stabilizes *Ifnb1* mRNA.**
**a**, Immunoblot analysis of transduced BMM immunoprecipitate (IP), stimulated with doxycycline and 100 μg ml⁻¹ DMXAA for 5–7 h (source data are provided in Supplementary Fig. 1). **b**, RT–qPCR analysis of *Ifnb1* from BMMs that were electroporated with the indicated Cas9–gRNA ribonucleoproteins (RNPs) with 8 h 100 μg ml⁻¹ DMXAA stimulation. The knockout efficiency was >85%. UT, untreated. **c**, RT–qPCR analysis of *Ifnb1* from BMMs with 8 h 100 μg ml⁻¹ DMXAA stimulation. **d**, BMM supernatant ELISA, 24 h 100 μg ml⁻¹ DMXAA stimulation. **e**, Roadblock RT–qPCR analysis of BMMs electroporated with the indicated Cas9–gRNA RNPs and treated with 100 μg ml⁻¹ DMXAA then 4SU. The knockout efficiency was 71% for *Resist1* and 51–69% for *Resist2*. The asterisks for timepoints, coloured by condition, indicate significance versus B6 + NTC. The black bars and asterisks indicate comparisons between *Sp140⁻/⁻* + NTC and *Sp140⁻/⁻* + *Resist1/2* gRNA. **f**, RT–qPCR analysis of *Ifnb1* from transduced BMMs treated with doxycycline and 100 μg ml⁻¹ DMXAA for 7 h. **g**, Roadblock

RT–qPCR analysis of *Ifnb1* from transduced B6 BMMs stimulated with doxycycline and 100 μg ml⁻¹ DMXAA, then 4SU at 2 h. The asterisks indicate significance versus mCherry. **h**, RT–qPCR analysis of *IFNB1* from transduced human BlaER1 monocytes stimulated with ADU-S100 and doxycycline. The asterisks indicate significance versus mCherry. **i**, Mouse lung colony-forming units (CFU) 96 h after *L. pneumophila* infection. Three independent pooled experiments; $n = 3$ wells (**b**–**h**). For **i**, the number of mice is indicated in the figure. Data are mean ± s.e.m. Statistical analysis was performed using one-way ANOVA with post hoc Dunnett's T3 multiple-comparison correction (**b**, **c** and **f**) or FDR correction (**d**), two-way ANOVA with post hoc Tukey's correction (**e**, **g** and **h**) and Kruskal–Wallis one-way ANOVA and Dunn's correction (**i**). Representative results are shown from four (**a**), three (**b** and **d**) and two (**c**, **e**, **f**, **g** and **h**) independent experiments. *$P < 0.05$, **$P < 0.005$, ***$P < 0.0005$, ****$P < 0.0001$; NS, not significant. Exact *P* values are provided in the Source data.

not likely to be a cell surface receptor as the DeepTMHMM algorithm[35] does not predict transmembrane helices. Moreover, recent work using unbiased mass spectrometry of immunoprecipitated human ANXA2R did not identify ANXA2 but, instead, identified the CCR4–NOT complex as the predominant ANXA2R-binding partner in cells[36]. We also found that a purified recombinant human ANXA2R protein did not robustly interact with the ANXA2–S100A complex, whereas a confirmed ANXA2 interaction partner (SMARCA3)[37] was able to bind to ANXA2–S100A (Extended Data Fig. 3c).

CCR4–NOT is the major mRNA deadenylase complex in eukaryotic cells[38]. Specific mRNAs targeted for deadenylation are recruited to CCR4–NOT through mRNA-binding proteins that use small peptide

motifs to dock to multiple interfaces on CCR4–NOT[38]. Recruited mRNAs are subsequently deadenylated, leading to their destabilization and degradation[38]. Human ANXA2R was proposed to inhibit CCR4–NOT and appeared to have antiviral effects[36], although the antiviral mechanism was unclear. As a unified hypothesis to explain how SP140 regulates *Ifnb1* mRNA stability and how ANXA2R provides antiviral defence, we propose that ANXA2R—which we now refer to as RESIST—stabilizes *Ifnb1* mRNA by binding to CCR4–NOT to inhibit *Ifnb1* mRNA recruitment, deadenylation and decay.

We first confirmed that HA-tagged mouse RESIST expressed in primary BMMs co-immunoprecipitated with CNOT1, the major scaffolding protein of CCR4–NOT (Fig. 3a). To test whether RESIST drives

elevated *Ifnb1* mRNA levels in *Sp140⁻/⁻* BMMs, we disrupted *Resist1/2* genes by Cas9–guide RNA (gRNA) electroporation of primary BMMs (Fig. 3b). *Resist1/2* disruption almost entirely eliminated the elevated *Ifnb1* transcripts observed in DMXAA-treated *Sp140⁻/⁻* BMMs (Fig. 3b). We also generated *Sp140⁻/⁻Resist1⁻/⁻Resist2⁻/⁻* mice using CRISPR–Cas9 (Extended Data Fig. 3d), and confirmed that RESIST drove elevated *Ifnb1* transcript levels and IFNβ protein levels in *Sp140⁻/⁻* BMMs (Fig. 3c,d). Analysis using Roadblock RT–qPCR confirmed that the decreased *Ifnb1* mRNA levels observed in *Resist1/2*-knockout cells were due to decreased mRNA stability (Fig. 3e). Furthermore, RESIST overexpression in primary BMMs was sufficient to elevate *Ifnb1* transcript levels more than tenfold, and eliminated the difference in *Ifnb1* transcript levels between B6 and *Sp140⁻/⁻* BMMs (Fig. 3f). RESIST overexpression also promoted *Ifnb1* transcript stabilization in primary BMMs, measured by Roadblock RT–qPCR (Fig. 3g). Overexpressed human RESIST also drove greater than tenfold elevated *IFNB1* transcript levels in human BlaER1 monocytes—which do not endogenously express RESIST—after stimulation with the STING agonist ADU-S100 (Fig. 3h). Finally, we found that *Resist1/2* entirely drove the susceptibility of *Sp140⁻/⁻* mice to *Legionella pneumophila* infection (Fig. 3i), which we have previously shown to be largely IFN-I dependent[4]. These results identify mouse and human RESIST as a potent positive regulator of IFN-I.

## Mechanism of RESIST

To elucidate the mechanism by which RESIST stabilizes *Ifnb1* mRNA, we used AlphaFold-Multimer to generate predicted structures of the interaction between mouse RESIST and CCR4–NOT subunits implicated in IFN-I regulation, including (1) CNOT10 and CNOT11 (CNOT10/11), which recruit mRNA-binding proteins and negatively regulate IFN-I signalling in T cells[39,40]; (2) the CNOT1 M-HEAT, which binds to tristetraprolin (TTP), an RBP that has been implicated in negative regulation of *Ifnb1* mRNA stability[41–46]; and (3) CNOT9, which acts as a secondary binding site for TTP[47] and was shown to directly bind to ROQUIN1, which, along with ROQUIN2, negatively regulates transcript stability for pro-inflammatory cytokines[10,48]. Our predicted structures suggested that RESIST may interact with multiple sites on the CCR4–NOT complex (Fig. 4a and Extended Data Fig. 4a). Such multivalent interactions are typical of many CCR4–NOT-interacting proteins[38]. Notably, the RESIST C-terminal region is predicted to fold into a helix (residues 168–177) and to interact with CNOT1 M-HEAT at the site that was previously shown to be bound by TTP[41] (Fig. 4a and Extended Data Fig. 4). Moreover, RESIST was predicted to bind to the tryptophan-binding pockets on CNOT9 that are also known to interact with TTP[47] (Fig. 4a and Extended Data Fig. 5). Both interactions were predicted with high confidence and were characterized by hydrophobic interactions between RESIST and CNOT1/9 (Extended Data Figs. 4b–e and 5a–d). AlphaFold also predicted, with lower confidence, that the RESIST C-terminal region interacted with a CNOT9 interface bound by *Drosophila* Roquin[48] (Fig. 4a and Extended Data Fig. 5a–d). By contrast, AlphaFold did not predict a high-confidence interaction between RESIST and CNOT10/11 (Extended Data Fig. 6).

To assess the functionality of CCR4–NOT subunits predicted to interact with RESIST, we genetically disrupted genes encoding CCR4–NOT subunits in BMMs with Cas9–gRNA electroporation. We evaluated *Ifnb1* transcript levels using RT–qPCR at the late 8 h timepoint after DMXAA stimulation (Fig. 4b). Notably, targeting of the CCR4–NOT scaffold gene *Cnot1* was not well-tolerated, with five- to tenfold fewer cells recovered after electroporation relative to non-targeting controls (NTCs), and variable knockout efficiency between experiments. Nevertheless, *Cnot1* deficiency phenocopied RESIST overexpression and led to similarly increased *Ifnb1* after DMXAA stimulation in both B6 and *Sp140⁻/⁻* cells (Fig. 4b and Extended Data Fig. 7a), consistent with the hypothesis that RESIST elevates *Ifnb1* transcript levels by inhibition of CCR4–NOT. By contrast, targeting *Cnot11* did not regulate *Ifnb1* in

either B6 or *Sp140⁻/⁻* BMMs (Fig. 4b and Extended Data Fig. 7a), demonstrating that RESIST-mediated *Ifnb1* mRNA stability was not regulated through CNOT11, consistent with the absence of a predicted interaction. Notably, deletion of *Cnot9* eliminated elevated *Ifnb1* transcripts in DMXAA-stimulated *Sp140⁻/⁻* BMMs, but did not affect *Ifnb1* levels in DMXAA-stimulated B6 BMMs (Fig. 4b and Extended Data Fig. 7a). These results indicate that CNOT9 is required for RESIST function. Using purified recombinant RESIST and CNOT1/9 proteins, we were able to observe a direct stoichiometric interaction between RESIST and CNOT1/9 (Fig. 4c).

To examine the functional importance of the RESIST C-terminal region, we generated a truncation mutant (RESIST(ΔC)). BMMs transduced with the RESIST(ΔC) mutant induced much lower levels of *Ifnb1* transcript than BMMs transduced with WT RESIST (Fig. 4d), suggesting that the C-terminal region is essential for RESIST-mediated *Ifnb1* mRNA stabilization. Notably, the RESIST(ΔC) mutant still immunoprecipitated with the CCR4–NOT complex (Fig. 4e), consistent with a predicted multivalent interaction between RESIST and the CCR4–NOT complex (Fig. 4a).

RESIST is predicted to interact with the known interfaces that TTP and ROQUIN bind to on the CCR4–NOT complex to mediate the decay of their target transcripts. We therefore examined whether TTP and/or ROQUIN mediate *Ifnb1* mRNA decay. We disrupted the genes encoding ROQUIN1/2 (*Rc3h1/2*), as well as TTP (*Zfp36*) alone and in combination with additional TTP family members (*Zfp36l1* and *Zfp36l2*), and evaluated *Ifnb1* transcript levels using RT–qPCR. Loss of *Zfp36* in DMXAA-stimulated B6 BMMs increased *Ifnb1* transcript levels (Fig. 4f and Extended Data Fig. 7b). By contrast, genetic disruption of *Rc3h1/2* did not affect *Ifnb1* induced by DMXAA in either B6 or *Sp140⁻/⁻* BMMs (Extended Data Fig. 7c,d). We validated that the RNA-binding zinc-finger domain of TTP was able to specifically bind to the *Ifnb1* 3′ untranslated region (3′-UTR) in a manner dependent on an AU-rich element (ARE) that is a canonical TTP target sequence (Extended Data Fig. 8). However, *Sp140* deficiency further increases *Ifnb1* in the absence of *Zfp36*, implying that RESIST has additional activities beyond TTP inhibition in *Sp140⁻/⁻* BMMs (Fig. 4f and Extended Data Fig. 7b). Consistent with this hypothesis, deletion of the genes encoding the additional TTP family members *Zfp36l1* and *Zfp36l2* resulted in further elevation of *Ifnb1* (Fig. 4f and Extended Data Fig. 7b). These results imply a previously unappreciated role for these TTP family members in the negative regulation of *Ifnb1* mRNA stability. Crucially, loss of *Sp140* does not affect *Ifnb1* levels in the absence of the *Zfp36* gene family (Fig. 4f). These results are consistent with a model in which RESIST interferes with the ability of TTP family members to mediate *Ifnb1* transcript destabilization. Indeed, we found that RESIST inhibited the interaction between TTP and CNOT1 when both RESIST and TTP were co-expressed in HEK293T cells (Fig. 4g). A potential model consistent with our data is that RESIST binds to CCR4–NOT and hinders the interaction between the CCR4–NOT complex and TTP family members, leading to *Ifnb1* mRNA stabilization (Fig. 4h). By contrast, CNOT9, which is required for elevated *Ifnb1* in *Sp140⁻/⁻* cells (Fig. 4b), appears to promote RESIST function, potentially by facilitating the interaction between RESIST and CCR4–NOT (Fig. 4h).

## Antiviral activity of SP140

Finally, we considered why SP140 may have evolved to repress the antiviral cytokine IFN-I. SP140 is part of the SP family that includes SP100, a well-described antiviral protein[3,49], and SP140L, which was recently described to have an antiviral role against the herpesvirus Epstein–Barr virus[50]. Mechanistically, SP100 is thought to inhibit transcription of viral genomes through sequestration in transcriptionally repressive NBs that co-localize with the NB protein PML[49]. Whether SP140 co-localizes with PML or has antiviral activity is unclear[3,51–53]. Importantly, because SP100 and other NBs are antiviral, viruses often encode effectors to

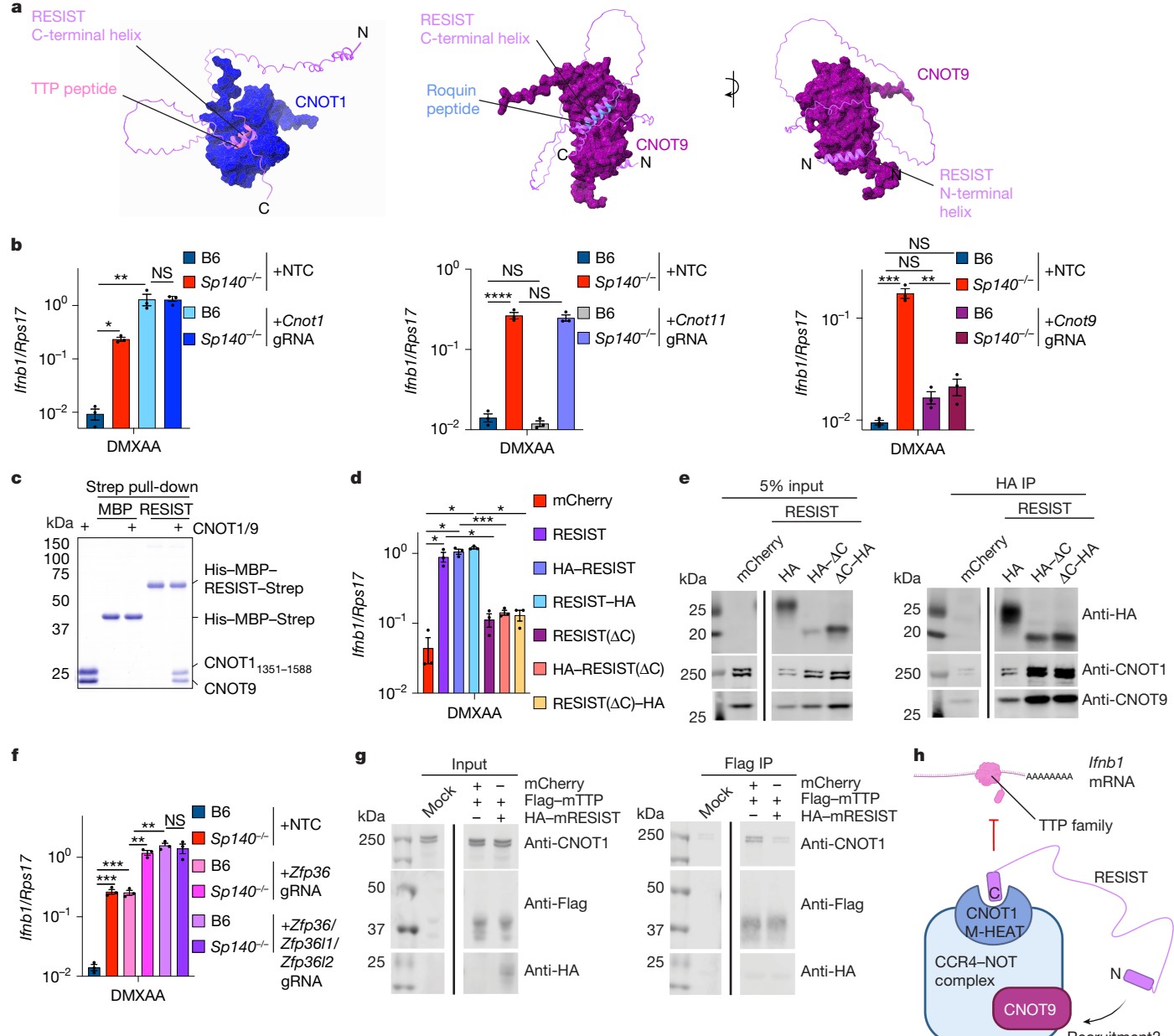

**Fig. 4 | RESIST counteracts repression of IFN-I by TTP-family proteins, a function that requires a RESIST C-terminal region and CNOT9. a**, AlphaFold predictions of RESIST with CNOT1 M-HEAT and CNOT9. TTP peptide–CNOT1 M-HEAT is from PDB 4J8S (ref. 41). ROQUIN peptide–CNOT9 is from PDB 5LSW (ref. 48). The N-terminal and C-terminal ends of RESIST are marked by N and C, respectively. **b**, RT–qPCR analysis of *Ifnb1* from BMMs electroporated with non-targeting control gRNA (NTC) or gRNAs targeting the indicated CCR4–NOT subunits after treatment for 8 h with 100 µg ml⁻¹ DMXAA. **c**, Strep pull-down of purified recombinant full-length human His–MBP–RESIST–Strep or His–MBP–Strep with human CNOT9 and CNOT1 (amino acids 1351–1588). The first lane indicates purified CNOT9 and CNOT1. **d**, RT–qPCR analysis of *Ifnb1* for BMMs transduced with the indicated lentiviral constructs and treated with doxycycline and DMXAA for 6 h. The results include data that are also shown in Fig. 3f. **e**, Immunoblot analysis of anti-HA IP of BMMs transduced with the indicated

constructs in **d**. The RESIST construct is C-terminally tagged with HA. Gel source data are provided in Supplementary Fig. 1. **f**, RT–qPCR analysis of *Ifnb1* from BMMs electroporated with the indicated gRNAs and treated for 8 h with 100 µg ml⁻¹ DMXAA. **g**, Immunoblot analysis of Flag IP of Flag–TTP (mouse) co-expressed with mouse HA–RESIST in HEK293T cells. Gel source data are provided in Supplementary Fig. 1. **h**, Schematic of how RESIST may interact with CCR4–NOT subunits CNOT1 and CNOT9 to mediate the stabilization of *Ifnb1* mRNAs. The diagram was created using BioRender.com. Data are mean ± s.e.m. Statistical analysis was performed using one-way ANOVA tests with post hoc Dunnett's T3 multiple-comparison correction. *n* = 3 wells of cells (**b**, **d** and **f**). Results are representative of two independent experiments (**b**–**g**). *$P < 0.05$, **$P < 0.005$, ***$P < 0.0005$, ****$P < 0.0001$; NS, not significant. Exact *P* values are provided in the Source data.

disrupt NB function[29,49,54]. To counter the effector-mediated disruption of NBs, we previously proposed that antiviral NB proteins can evolve the ability to repress IFN-I as a secondary function[5,29]. Viruses that disrupt NBs to evade their primary antiviral activity would thereby unleash a secondary 'backup' interferon response. As the secondary response is elicited by a pathogen effector, it is classified as effector-triggered

immunity—a major immune strategy in plants[6] that remains poorly characterized in mammals[5].

To test the hypothesis that SP140 is an antiviral NB protein, we first characterized the nature of SP140 NBs. To detect endogenously expressed SP140, we generated *HA-Sp140* knock-in mice by inserting an HA-tag after the start codon of the *Sp140* gene (Extended Data Fig. 9a).

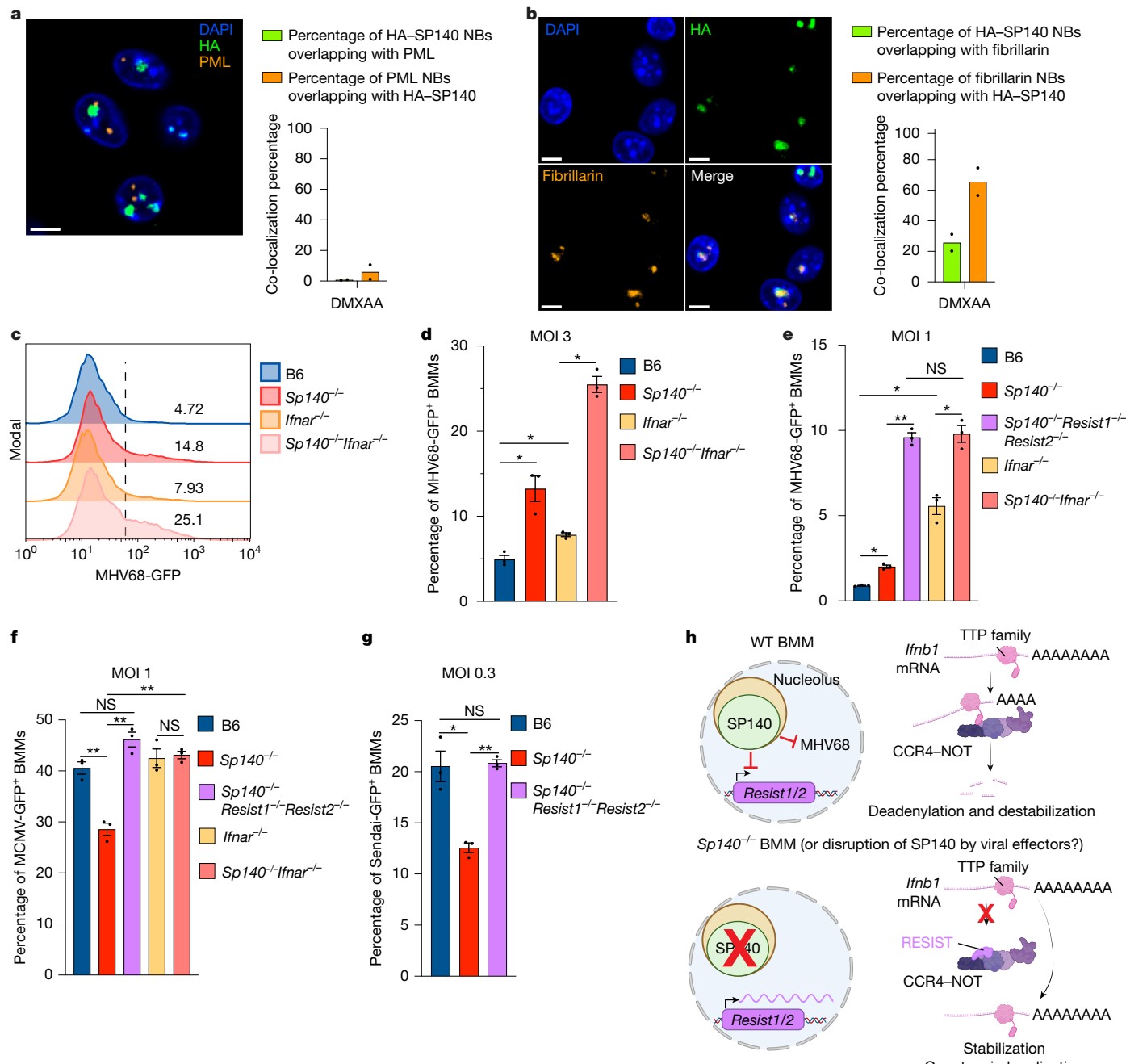

**Fig. 5 | SP140 is an antiviral NB protein that co-localizes with nucleoli.**
**a**, Immunofluorescence analysis of DMXAA-treated *HA-Sp140*[+/+] BMMs, with DAPI, anti-HA and anti-PML staining. HA–SP140 and PML NB overlap was quantified for two independent experiments. Scale bar, 5 μm. **b**, Immunofluorescence analysis of DMXAA-treated *HA-Sp140*[+/+] BMMs stained with DAPI, anti-HA and anti-fibrillarin. HA–SP140 and fibrillarin NB overlap was quantified for two independent experiments. Scale bars, 5 μm. **c**, The MHV68-GFP signal in MHV68-GFP-infected BMMs was assessed using flow cytometry. The numbers represent MHV68-GFP[+] as a percentage of live cells. **d**, Quantification of MHV68-GFP[+] cells from **c**. *P* = 0.0225 (B6 versus *Sp140*[−/−]), *P* = 0.0161 (B6 versus *Ifnar*[−/−]) and *P* = 0.0073 (*Ifnar*[−/−] versus *Sp140*[−/−]*Ifnar*[−/−]). MOI, multiplicity of infection. **e**, Quantification of MHV68-GFP[+] BMMs. *P* = 0.007 (B6 versus *Sp140*[−/−]), *P* = 0.0025 (*Sp140*[−/−] versus *Sp140*[−/−]*Resist1*[−/−]*Resist2*[−/−]), *P* = 0.0136 (B6 versus *Ifnar*[−/−]) and

*P* = 0.007 (*Ifnar*[−/−] versus *Sp140*[−/−]*Ifnar*[−/−]). **f**, Quantification of MCMV-GFP[+] BMMs, assessed using flow cytometry. *P* = 0.0042 (B6 versus *Sp140*[−/−]), *P* = 0.0035 (*Sp140*[−/−] versus *Sp140*[−/−]*Resist1*[−/−]*Resist2*[−/−]) and *P* = 0.0035 (*Ifnar*[−/−] versus *Sp140*[−/−]*Ifnar*[−/−]). **g**, Quantification of Sendai-GFP[+] BMMs, assessed using flow cytometry. *P* = 0.0251 (B6 versus *Sp140*[−/−]) and *P* = 0.0006 (*Sp140*[−/−] versus *Sp140*[−/−]*Resist1*[−/−]*Resist2*[−/−]). **h**, Schematic of the proposed model of SP140 antiviral activity and RESIST-mediated *Ifnb1* transcript stabilization. The diagram was created using BioRender.com. Data are mean ± s.e.m. *n* = 3 wells (**d**–**g**). Statistical analysis was performed using one-way ANOVA with FDR correction (**d**–**g**). Results are representative of two (**a** and **b**) or three (**c**–**g**) independent experiments. **P* < 0.05, ***P* < 0.005; NS, not significant. Exact *P* values are provided in the Source data.

Immunoblot analysis of BMMs from *HA-Sp140*[+/+] mice confirmed the presence of an anti-HA reactive band at the expected molecular mass of SP140 (Extended Data Fig. 9b). HA–SP140 was expressed at similar levels to WT SP140, and was functional, as assessed by its ability to

repress *Ifnb1* at late timepoints in DMXAA-treated BMMs (Extended Data Fig. 9b,c). Notably, immunofluorescence analysis of *HA-Sp140*[+/+] BMMs demonstrated that SP140 forms large NBs that do not co-localize with PML bodies (Fig. 5a and Extended Data Fig. 9d). Rather, we found

that SP140 NBs appear to partially colocalize with the nucleolar marker fibrillarin (Fig. 5b and Extended Data Fig. 9e).

We next tested whether SP140 has antiviral activity. We infected B6 and $Sp140^{-/-}$ BMMs with MHV68-GFP, a mouse herpesvirus engineered to encode GFP. We found that a larger fraction of $Sp140^{-/-}$ BMMs was GFP positive compared with WT B6 BMMs (Fig. 5c,d and Extended Data Fig. 10a). The antiviral effect of SP140 was independent of IFN-I signalling, as $Sp140^{-/-}Ifnar^{-/-}$ BMMs also exhibited higher levels of infection compared with $Ifnar^{-/-}$ BMMs (Fig. 5c,d). We also observed that $Sp140^{-/-}$ BMMs exhibited higher $Ifnb1$ transcript levels than WT cells after MHV68-GFP infection (Extended Data Fig. 10b). Consistent with this observation and the role of RESIST as a critical driver of IFN-I in $Sp140^{-/-}$ BMMs, $Sp140^{-/-}Resist1^{-/-}Resist2^{-/-}$ BMMs were more susceptible to MHV68-GFP infection than $Sp140^{-/-}$ BMMs, and phenocopied the susceptibility of $Sp140^{-/-}Ifnar^{-/-}$ BMMs (Fig. 5e). These data suggest that elevated RESIST-dependent IFN-I in $Sp140^{-/-}$ BMMs counteracts the loss of the antiviral protein SP140 during MHV68-GFP infection.

We next tested whether SP140 has antiviral activity against other viruses. We infected B6 and $Sp140^{-/-}$ BMMs with MCMV-GFP, another mouse herpesvirus engineered to encode GFP. In contrast to MHV68, we found that MCMV-GFP was modestly restricted in $Sp140^{-/-}$ BMMs (Fig. 5f). The restriction of MCMV-GFP in $Sp140^{-/-}$ cells depended on RESIST and IFNAR (Fig. 5f). We also found that $Sp140^{-/-}$ BMMs restricted the replication of an RNA virus, Sendai virus, also encoding GFP (Fig. 5g), whereas $Sp140^{-/-}Resist1^{-/-}Resist2^{-/-}$ BMMs and WT BMMs exhibited similar levels of Sendai-GFP replication. These results suggest the antiviral activity of SP140 is specific to certain viruses such as MHV68-GFP, and that the elevated IFN-I due to RESIST expression in the absence of SP140 counters replication of diverse viruses.

## Discussion

Our results describe several findings. First, we identify RESIST as a positive regulator of IFN-I that operates through promoting $Ifnb1$ mRNA stability. We propose that RESIST may represent a founding example of a cytokine regulator that functions through CCR4–NOT inhibition. Second, we demonstrate that RESIST is a direct target of SP140 transcriptional repression in mouse macrophages, and that RESIST enhances $IFNB1$ mRNA stability in both mouse and human cells. Third, we demonstrate that the RNA-binding protein TTP (ZFP36) as well as its previously poorly described paralogues ZFP36L1 and ZFP36L2 are important but partly redundant negative regulators of IFN-I. Lastly, we describe an antiviral function for SP140 against MHV68 that is independent of its ability to regulate IFN-I.

At present, the mechanism by which SP140 is antiviral is unclear. SP140 contains domains homologous to those found in the antiviral protein SP100. As in SP100, the DNA-binding SAND domain of SP140 may also bind to viral genomes, while the caspase-activation and recruitment domain may mediate SP140 oligomerization into transcriptionally repressive NBs[49]. Notably, our results suggest that SP140 localizes distinctly from SP100 and PML, suggesting the antiviral mechanism of SP140 may be distinct from that of SP100. How SP140 is antiviral will be of great interest to determine in future work.

Our data support a model in which RESIST counteracts destabilization of $Ifnb1$ mRNA mediated by CCR4–NOT and the TTP family. The activity of RESIST requires its C-terminal region. We predict that RESIST interacts through multiple binding sites with the CCR4–NOT complex. Notably, the predicted sites of RESIST interaction overlap with known sites of interaction between CCR4–NOT and TTP. Validation of the predicted interactions between RESIST and CCR4–NOT, and a detailed investigation of whether RESIST competes with TTP family members for CCR4–NOT interaction, will require further work, including structural analyses of the RESIST–CCR4–NOT complex.

Our data do not exclude the possibility that RESIST acts as a general inhibitor of CCR4–NOT, similar to RNF219 (ref. 55). However, in $Sp140$-deficient cells, we observe few upregulated transcripts other than $Resist1$ and $Ifnb1$ (Fig. 2a), suggesting that RESIST acts to stabilize a specific set of mRNA targets. Notably, we also found that known TTP-regulated transcripts such as $Tnf$ and $Il6$ are not derepressed in $Sp140^{-/-}$ macrophages (Fig. 2a). The apparent specificity of RESIST could arise from multiple mechanisms, including cellular co-localization of RESIST and $Ifnb1$ mRNA transcripts, preferential targeting of $Ifnb1$ mRNA by the TTP family members or particular sensitivity of $Ifnb1$ transcripts to the partial TTP inhibition mediated by RESIST. It is also possible that RESIST may regulate other transcripts besides $Ifnb1$ in vivo. Ultimately, further work is required to determine the mechanistic basis for the apparent specificity of RESIST for $Ifnb1$ in macrophages, and whether RESIST regulates other transcripts in vivo.

While this work focuses on the SP140–RESIST circuit, RESIST may also function independently of SP140. In human cells, RESIST is induced by IFN-I[36], which suggests that it may have an antiviral function. It remains unclear whether human SP140 or other SP family members also regulate human $RESIST$. In WT SP140-sufficient mouse macrophages, we do not observe a role for RESIST, presumably because SP140 strongly represses RESIST expression. However, it is possible that RESIST is regulated differently in other cell types in vivo.

Our results highlight an important role for TTP and TTP family members ZFP36L1 and ZFP36L2 in IFN-I repression in macrophages. Our finding that TTP represses IFN-I is consistent with previous work that suggests that TTP negatively regulates the stability of $Ifnb1$ mRNA[42,43,45,46]. TTP detectably binds to the 3′-UTR of $Ifnb1$ mRNA[42] (Extended Data Fig. 8), and a hyperactive TTP mutant represses $Ifnb1$ transcript levels in mouse macrophages[43]. ZFP36L1 and ZFP36L2 are understudied, and their genetic deletion in mice is either embryonically or perinatally lethal[44]. However, ZFP36L1 and ZFP36L2 recruit CCR4–NOT to target mRNAs and modulate lymphocyte function[44,56–58]. Notably, while $Zfp36^{-/-}$ mice develop severe autoinflammation, mice lacking myeloid-specific TTP expression do not develop immunopathology[44], suggesting functional redundancy among TTP family members in myeloid cells. Indeed, recent research described fatal autoinflammation in mice lacking myeloid-specific expression of $Zfp36$, $Zfp36l1$ and $Zfp36l2$ that may result partly from elevated $Ifnb1$ transcripts in these mice[46].

Our results confirm that SP140 resides in NBs and show that it is intrinsically antiviral against MHV68 (Fig. 5d,e). Moreover, we show that SP140 has evolved a secondary function−repression of the $Resist1/2$ genes, which encode a positive regulator of IFN-I. We propose that these apparently conflicting anti- and pro-viral roles of SP140 can be explained by a host−virus evolutionary arms race in the nucleus (Fig. 5h). Many viral effectors target antiviral NBs[29,49,54]. We speculate that SP140 evolved to protect itself from viral attack by acquiring the ability to negatively regulate RESIST. In this scenario, viral attack of SP140 would de-repress RESIST, which would augment IFN-I-mediated antiviral defence. Thus, the SP140–RESIST pathway may provide an example of effector-triggered immunity[5,6,29].

Finally, we speculate that the antiviral activity of SP140 could explain why multiple sclerosis and B cell cancers are linked to $SP140$ mutations, as infection with MHV68-related viruses Epstein−Barr virus and Kaposi's sarcoma-associated herpesvirus are associated with these immune disorders[59,60]. In future work, it will be of interest to determine whether individuals with $SP140$ mutations are more susceptible to infection with Epstein−Barr virus and Kaposi's sarcoma-associated herpesvirus, which could lead to increased risk of multiple sclerosis and B cell cancers.

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

# Methods

## Mice

All animal experiments complied with the regulatory standards of, and were approved by, the University of California Berkeley Institutional Animal Care and Use Committee. Mice were maintained in specific-pathogen-free conditions under a 12 h–12 h light–dark cycle at 20–26 °C and 30–70% humidity, and given water and standard chow diet (Harlan's irradiated chow) ad libitum. All mice were bred in-house. C57BL/6 and *B6.129S2-Ifnar1*[tm1Agt]/*Mmjax* (*Ifnar*[−/−]) mice were originally from Jackson Laboratories and MMRC and further bred in-house. The generation and genotyping of *Sp140*[−/−] mice was previously described[4]. *HA-Sp140* knock-in mice were generated by electroporation of C57BL/6 zygotes with Cas9, Alt-R CRISPR-Cas9 crRNA from IDT (gRNA sequence: ACUCCAAGGGACCCUGUUCA), and a homology repair Alt-R HDR donor oligo from IDT (CCCCTGAAGGAGTTCTCTCTGGGCTTCCCAGAGAC TCAGAGGGGGTTCGGTCTAGTCTGAACAGGGTCCCTTGGAGTCTGTGT AGGGGATGTACCCATACGATGTTCCAGATTACGCTGCAGGAGGCTACA ATGAACTCAGCAGCAGGTAAGTCCCATTCTCTCTTGTCCCTTGTCTC) as previously described[61]. Founders were backcrossed to C57BL/6J, and mice with matching *HA-Sp140* alleles were further bred. *HA-Sp140* knock-in mice were genotyped using a qPCR-based assay from Transnetyx. *Sp140*[−/−]*Resist1*[−/−]*Resist2*[−/−] mice were generated by electroporation of *Sp140*[−/−] zygotes with Cas9 and sgRNA (CGACGATGGCGGTGACT ACC)[4]. Founders were genotyped and backcrossed to *Sp140*[−/−] mice, and progeny with matching alleles were further bred. For genotyping *Resist1/*2, large ear clips were obtained and digested in QuickExtract lysis buffer overnight with 0.4 mg ml[−1] proteinase K, followed by heat inactivation (85 °C for 4 min, 98 °C for 2 min) and stored at −80 °C. After freezing, ear clip lysates were vortexed. Mice were genotyped for *Resist1* (*Gm21188*) with Q5 2× PCR mix according to the manufacturer's instructions with 1 µl of lysate per 20 µl PCR reaction (F, TTGAG AAATCCGTTTGTAATGGG; R, GCCTTTCTCCGGATTCACGA; cycling conditions: 98 °C for 3 min, 35 cycles of 98 °C for 10 s, 63.5 °C for 20 s and 72 °C for 65 s, followed by a final extension of 72 °C for 10 min). Mice were genotyped for *Resist2* (*Gm36079*) with Phusion GC rich PCR components according to the manufacturer's instructions, using 1 µl of lysate per 20 µl PCR reaction (F, TGGTATTCTCTAGAGATAACATCAC AGCACCTACTTACTCC; R, CCTCCCCTCGCCATCACTGCCTG; cycling conditions: 98 °C for 30 s, 30 cycles of 98 °C for 10 s, 72 °C for 15 s and 72 °C for 60 s, followed by a final extension of 72 °C for 10 min). A total of 5 µl of PCR product was cleaned with FastAP and ExoI, then diluted twofold with water and Sanger sequenced *(Resist1* R, GCCTTTCTCCGG ATTCACGA; *Resist2* F seq, CTGAATGATTCTTCTACTGCTTCCATCC). Sanger sequencing results were evaluated with Snapgene (v.7.0.1).

## Cell culture

HEK293T and GP2 cells were obtained from the UC Berkeley Tissue Culture facility and further propagated at 37 °C and 5% CO$_2$ in complete DMEM (10% FBS (v/v) (Gibco) and 1× penicillin–streptomycin and glutamine (Gibco)). BlaER1 cells (from the laboratory of V. Hornung) were cultured in the B cell stage in complete RPMI, and differentiated into monocytes as described previously[29]. Cell lines tested negative for mycoplasma by PCR (F, CACCATCTGTCACTCTGTTAACC; R, GGAGC AAACAGGATTAGATACCC) with Dreamtaq PCR reagents and validated with short-tandem-repeat profiling by the UC Berkeley Tissue culture facility. HEK293T cells were last validated on 25 November 2024, GP2 cells were last validated on 23 May 2024 and BlaER1 cells were last validated on 23 May 2024.

## BMM generation and stimulation

Mice were euthanized and bones (femurs and tibias) were extracted with stringent washing in 70% ethanol. After washing in 70% ethanol and BMM medium (complete RPMI with 10% MCSF (v/v) generated from 3T3 cells as described previously[4]), bones were crushed in a sterilized mortar and pestle, and bone marrow was passed through a 70-µm filter. Bone marrow from one mouse was divided across eight 15 cm non-tissue-culture-treated plates in 30 ml total volume in BMM medium. The day of bone marrow collection was considered to be day 0, and BMMs were fed on day 3 with 10 ml medium per plate. On day 6, BMMs were collected in cold PBS by scraping and seeded onto non-tissue-culture-treated plates at the appropriate density (100,000 cells per non-tissue-culture-treated 12-well and 24-well; 6-well for over 100,000 cells per well) and rested at least overnight before stimulation. To stimulate, the medium was aspirated and replaced with BMM medium containing 10 ng ml[−1] LPS (Invivogen, tlrl-3pelps), 100 µg ml[−1] poly(I:C) (Invivogen, tlrl-picw), 100 µg ml[−1] DMXAA (Cayman Chemicals, 14617) or 10 ng ml[−1] IFNγ (BioLegend, 575304).

## RT–qPCR

RNA was isolated using the Omega Biotek Total RNA II kit according to the kit instructions. RNA was subsequently DNase-treated either on-column (Qiagen, 79254) or with RQ1 (Promega, M6101) according to the manufacturer's instructions. For BlaER1 cells, RNase inhibitor was always included with DNase treatment (either RNaseOUT (Invitrogen) or RNasin (Promega)). RNA was then converted to cDNA with Superscript III Reverse Transcriptase (Invitrogen, 18080093) and oligo dT$_{18}$ (NEB, S1316S) in the presence of RNase inhibitors. Diluted cDNA was assessed by qPCR using the Power SYBR Green PCR Master Mix (Thermo Fisher Scientific, 43-676-59) reagents, using technical duplicates. A standard curve generated from samples within each experiment was used to quantify relative amounts of transcript. *Ifnb1* transcript levels (F, GTCCTCAACTGCTCTCCACT; R, CCTGCAACCACC ACTCATTC) were normalized to housekeeping genes including *Rps17* (F, CGCCATTATCCCCAGCAAG; R, TGTCGGGATCCACCTCAATG), *Oaz1* (F, GTGGTGGCCTCTACATCGAG; R, AGCAGATGAAAACGTGGTCAG) or *Hprt1* (F, GTTGGATACAGGCCAGACTTTGTTG; R, GAGGGTAGGCTGGCC TATAGGCT) as indicated in the figures. Human *IFNB1* transcript quantities (F, CAGCATCTGCTGGTTGAAGA; R, CATTACCTGAAGGCCAAGGA) were normalized to *HPRT1* (F, ATCAGACTGAAGAGCTATTGTAATGA; R, TGGCTTATATCCAACACTTCGTG). DNase-treated RNA that was not treated with reverse transcriptase was also included in qPCR assays as a control to ensure complete digestion of genomic DNA. RT–qPCR reaction wells with a poor ROX reference were excluded from analysis, as were samples with unusually low housekeeping gene amounts indicative of RNA degradation. Replicates in figures indicate biological replicates (separate wells of cells).

## Roadblock RT–qPCR

BMMs were either pretreated with freshly prepared 400 µM 4SU (Cayman Chemicals, 16373) 1–2 h before 100 µg ml[−1] DMXAA stimulation or treated with 4SU 2 h after DMXAA stimulation, as previously described[27,62]. In brief, RNA was collected as described above and DNase treated. Equal amounts of RNA (~200 ng or more) were treated with 48 µM *N*-ethyl maleimide (Sigma-Aldrich, 04259-5G) and quenched with 20 mM DTT. RNA was then purified with RNAClean beads (Beckman Coulter, A63987) and converted to cDNA with ProtoScript II Reverse Transcriptase according to the manufacturer's instructions in the presence of RNase inhibitors. qPCR was then performed as described above with the primers for *Ifnb1* (F, TGGATGGCAAAGGCAGTGTAA; R, CACCTACAGGGCGGACTTC) and *Rps17* (see above). In some experiments, housekeeping-normalized *Ifnb1* is displayed in the figures as the percentage of average *Ifnb1*/*Rps17* quantity present at 2 h of DMXAA stimulation for each condition. To validate the Roadblock RT–qPCR protocol, in every Roadblock RT–qPCR experiment, BMM cells were also pretreated in parallel with 4SU before DMXAA treatment, and RNA was isolated at $T = 2$ and other timepoints indicated in the figures and converted to cDNA. NEM treatment was confirmed to reduce the detection of *Ifnb1* transcripts by qPCR by around tenfold with 4SU pretreatment before DMXAA for every Roadblock RT–qPCR experiment.

## ELISA

BMMs were seeded at 85,000 cells per well in non-tissue-culture-treated 96-well plates in 200 µl of medium and rested overnight. Cells were stimulated with stimuli indicated in the figure legends for 24 h. The plates were spun at 600$g$ for 5 min, and the supernatants were removed and stored at −80 °C. Culture supernatants from DMXAA-treated cells were diluted 1:50–1:100 before evaluation using the Lumikine Xpress mIFN-β 2.0 kit according to the manufacturer's instructions.

## RNA-seq sample generation and analysis

Day 6 BMMs derived from B6 and $Sp140^{-/-}$, or $Ifnar^{-/-}$ and $Sp140^{-/-}Ifnar^{-/-}$ mice were seeded at $1 \times 10^6$ cells per well in six-well non-tissue-culture-treated plates and rested overnight. BMMs were then either left unstimulated or stimulated with DMXAA (100 µg ml$^{-1}$ for B6 and $Sp140^{-/-}$ BMMs, and 10 µg ml$^{-1}$ DMXAA for $Ifnar^{-/-}$ and $Sp140^{-/-}Ifnar^{-/-}$ BMMs) for either 8 h for B6 and $Sp140^{-/-}$ BMMs or for 4 h for $Ifnar^{-/-}$ and $Sp140^{-/-}Ifnar^{-/-}$ BMMs. RNA was isolated using the Omega Biotek Total RNA II kit, and DNase-treated with TURBO DNase (Thermo Fisher Scientific, AM2238) according to the manufacturer's instructions. For RNA from B6 and $Sp140^{-/-}$ BMMs, single-index libraries were generated by Azenta, using rRNA depletion (Qiagen, QIAseq FastSelect–HMR rRNA Removal kit) and the NEBNext RNA Ultra kit (NEB). Libraries were evenly split across two lanes of an Illumina HiSeq flow cell and sequenced (150 bp paired-end reads, depth of 25–30 million reads per sample). For $Ifnar^{-/-}$ and $Sp140^{-/-}Ifnar^{-/-}$ BMM samples, libraries were prepared for Illumina sequencing with dual indices using poly(A) selection and KAPA HyperPrep reagents (Roche) after heat fragmentation through the UC Berkeley QB3 Vincent J. Coates Genomics Sequencing Laboratory. Libraries were subsequently size-selected between 450 and 500 bp, and then sequenced on the Illumina NovaSeq flow cell for a depth of over 20 million mapped reads (150 bp, paired-end). B6 and $Sp140^{-/-}$ samples were collected separately from $Ifnar^{-/-}$ and $Sp140^{-/-}Ifnar^{-/-}$ samples. Three biological replicates for each genotype and condition were collected across three independent experiments from three separate mice for each genotype, which were age and sex-matched. Adapters and low-quality reads were trimmed using BBDuk v.38.05 with arguments 'ktrim=r k=23 mink=11 hdist=1 mapq=10 qtrim=r trimq=10 tpe tbo'. For UCSC genome browser visualization, reads were mapped to the mm10 mouse reference genome (https://genome.ucsc.edu/cgi-bin/hgGateway?db=mm10) using hisat2 v.2.1.0 with the options '--no-softclip -k 100 | samtools view -q 10 -Sb - | samtools sort'. CPM normalized bigwigs were made using deepTools bamCoverage v.3.0.1. For transcript quantification, reads were mapped to mm10 (gencode.vM18.annotation.gtf) using Salmon v.0.13.1 with the options '--libType A --validateMappings --rangeFactorizationBins 4 --gcBias'. DEGs were called using DESeq2 v.1.38.3 with design '-batch + genotype + treatment + genotype:treatment'. For differential expression analysis of $Sp140^{-/-}$ versus B6 BMMs and $Sp140^{-/-}Ifnar^{-/-}$ versus $Ifnar^{-/-}$ BMMs, normalized count data were derived from the following DESeq2 comparisons: (1) SP140-deficient BMMs treated with DMXAA (three replicates) versus SP140-WT BMMs treated with DMXAA (3 replicates); and (2) untreated SP140-deficient BMMs (3 replicates) versus untreated SP140-WT BMMs (3 replicates). Genes with zero counts across all samples were removed. Volcano plots were generated with ggplot2 v.3.5.0 R package.

## ATAC–seq sample generation and analysis

B6 and $Sp140^{-/-}$ samples BMMs were derived and stimulated as described above with 100 µg ml$^{-1}$ DMXAA for 8 h in three separate experiments, in parallel to samples generated for RNA-seq. BMMs were collected in PBS and counted, and ATAC–seq samples were generated from 100,000 input cells essentially as described previously[63], except that isolated nuclei were centrifuged at 1,000$g$. Illumina-compatible libraries were prepared as described previously[63], with additional Ampure XP bead (Beckman Coulter) purification to remove contaminating adaptor dimers. The samples were sequenced with Azenta on the Illumina HiSeq flow cell (over 50 million paired end reads per sample, 150 bp reads). Adapters and low-quality reads were trimmed using BBDuk v.38.05 with the arguments 'ktrim=r k=23 mink=11 hdist=1 maq=10 qtrim=r trimq=10 tpe tbo' and mapped to mm10 using BWA-MEM v.0.7.15, and only uniquely mapping reads with a minimum MAPQ of 10 were retained. Fragments aligning to the mitochondrial genome were removed. Peak calling was performed using complete and size-subsetted alignment files with MACS2 v.2.1.1 with paired-end options '--format BAMPE --SPMR -B --broad'. For visualization, counts-per-million-normalized bigwig files were made using deepTools bamCoverage v.3.0.1. For differential expression analysis of ATAC–seq peak data of $Sp140^{-/-}$ BMMs treated with DMXAA (3 replicates) versus B6 BMMs treated with DMXAA (3 replicates), normalized count data were derived from DESeq2. The nearest gene was found to the resultant peaks in data generated from $Sp140^{-/-}$ BMMs using closestBed (v.2.28.0). This gene list was then overlapped with the closestBed gene list of HA–SP140 CUT&RUN peak data. Volcano plots were generated with ggplot2 v.3.5.0 R package.

## BMM transduction

Low-passage HEK293T cells or GP2 packaging cells were seeded in a six-well format on tissue-culture-treated plates and rested at least overnight. To generate lentivirus, HEK293T cells at >70% confluency in a six-well tissue-culture-treated plate were transfected with 0.468 µg VSV-G (pMD2.G, Addgene, 12259), 1.17 µg D8.9 packaging vector and 1.56 µg of doxycycline-inducible, puromycin-selectable lentiviral vector per well using Lipofectamine 2000 according to the manufacturer's instructions. The lentiviral backbone used in this study (pLIP) was adapted from pLIX (Addgene, 41394) by removal of ccdB[29]. Gene blocks encoding codon-optimized mouse/human RESIST, or mCherry were cloned into pLIP, digested with NheI and BamHI, after the dox-inducible promoter, using Infusion reagents (Takara), essentially as described previously[29]. RESIST(ΔC) constructs were also generated by PCR to remove residues after Asp161 followed by Infusion cloning into the pLIP backbone. All of the constructs were validated by Sanger sequencing, the results of which were evaluated in Snapgene v.7.0.1. To generate retrovirus, GP2s at higher than 70% confluency in a six-well plate were transfected with 0.5 µg of VSV-G and 3.5 µg of retroviral vector per well with Lipofectamine 2000. Retroviral vectors in this study (SINV HA-SP140, SINV-SP140) were derived from the self-inactivating retrovirus pTGMP (Addgene, 32716; from the laboratory of S. Lowe). Mouse $Sp140$ or $HA$-$Sp140$ codon-optimized cDNA was cloned into pTGMP with Infusion (Takara), modified to include a minimal CMV promoter driving SP140 constructs, followed by a PGK promoter driving a puromycin-resistance cassette. Then, 18–20 h after transfection, medium on transfected cells was changed to 1 ml BMM medium. Bone marrow was collected as described above, and plated in BMM medium without dilution. Retronectin-treated (Takara) six-well plates were generated according to the manufacturer's instructions. The next day (around 30 h after changing medium on transfected cells), virus was collected from transfected cells by filtration of supernatant through a 0.45 µm filter and added to $1 \times 10^6$ cells per well of bone marrow in 4 ml total of BMM medium. Plates were spun at 650$g$ for 1.5–2 h at 37 °C. Then, 3 days after bone marrow collection, BMMs were fed with 1.33 ml BMM medium. BMMs were puromycin selected on day 4 after BMM collection with 2.75–5 µg ml$^{-1}$ puromycin. Puromycin kill curves were determined for every stock to identify the lowest concentration needed for BMM selection, and a non-transduced well or BMMs transduced with a retroviral vector lacking a puromycin resistance cassette were used to verify complete killing of non-transduced cells by puromycin. After puromycin selection (2 days), the medium was exchanged and BMMs were allowed to recover for 2–6 days before seeding. BMMs transduced with lentiviral pLIP constructs were pretreated overnight for 24 h with 2.5 µg ml$^{-1}$ doxycycline, then restimulated with 100 µg ml$^{-1}$ DMXAA and fresh doxycycline. Cells were collected at the timepoints indicated in the figure legends for either RNA isolation or co-IP.

## HA–SP140 CUT&RUN and analysis

*Sp140*[−/−] BMMs were transduced with retrovirus encoding HA–SP140 or SP140 as described above, and stimulated for 8 h with 100 µg ml[−1] DMXAA. Half a million cells were input into CUT&RUN[64] in biological triplicates using the Epicypher CUTANA ChIC/CUT&RUN kit (Epicypher, 14-1048, v3), using 0.5 µg of rabbit anti-HA monoclonal antibody (Cell Signaling Technologies, C29F4) with *Escherichia coli* genomic DNA spike-in. Non-transduced B6 BMMs (0.5 × 10[6]) were also processed for CUT&RUN with 0.5 µg of rabbit isotype control IgG (Epicypher, 13-0042), with a single biological replicate. CUT&RUN was carried out on isolated nuclei according to the kit instructions. Libraries were prepared using the Epicypher CUTANA CUT&RUN library prep kit (Epicypher, 14-1001) according to the kit instructions, then sequenced on the Illumina NovaSeq flow cell with 250 bp paired-end reads for around 6 million reads per sample. Adapters and low-quality reads were trimmed using BBDuk v.38.05 using the options 'ktrim=r k=23 mink=11 hdist=1 maq=10 tpe tbo qtrim=r trimq=10'. Trimmed reads were aligned to the mm10 assembly using BWA-MEM v.0.7.15, and only uniquely mapping reads with a minimum MAPQ of 10 were retained. Fragments aligning to the mitochondrial genome were removed. Peak calling was performed using complete and size subsetted alignment files with MACS2 v.2.1.1 with the paired-end options '--format BAMPE --pvalue 0.01 --SPMR -B --call-summits'. Bigwig files were prepared from the MACS2-normalized bedgraph files using bedGraphToBigWig v.4. MACS2 peak scores, the normalized number of sequence reads that originate from a bound genomic location, were output for HA–SP140 peaks.

## Cistrome and GREAT analysis

Replicate MACS2 CUT&RUN peak files were merged, then controls (IgG and SP140) were subtracted from the HA–SP140 peak file using bedtools intersect (v.2.28.0)[65] to output a file of SP140 peaks. This SP140 peak file was used as input in the cistrome toolkit data browser (http://dbtoolkit.cistrome.org/)[66] looking for significant binding overlap of histone marks and variants in mm10. The SP140 peak file was also used as input for the Genomic Regions Enrichment of Annotations Tool (GREAT, v.4.0.4)[67].

## Recombinant protein expression and purification

ANXA2-S100A was produced and purified in *E. coli* BL21 (DE3) Star cells (Thermo Fisher Scientific) in LB medium at 20 °C as a fusion protein carrying an N-terminal His6–SUMO tag. Cells were resuspended in lysis buffer (50 mM HEPES, 500 mM NaCl, 25 mM imidazole, pH 7.5) and lysed using the Branson Ultrasonics Sonifier SFX550. The lysate was cleared by centrifugation at 40,000*g* for 1 h at 4 °C. The cleared lysates were loaded onto the 5 ml HisTrap column (Cytiva). The bound protein was eluted over a linear gradient with elution buffer (50 mM HEPES, 200 mM NaCl, 500 mM imidazole, pH 7.5). For the final step, size-exclusion chromatography was performed on the Superdex 200 26/600 column in a buffer containing 10 mM HEPES, 200 mM NaCl, 2 mM DTT, pH 7.5.

For expression of full-length RESIST, full-length human RESIST (UniProt: Q3ZCQ2) was inserted between the BamHI and XbaI restriction sites of the pLIB plasmid[68] with a TEV (tobacco etch virus) protease-cleavable, N-terminal His6–MBP (maltose-binding protein) tag and a C-terminal StrepII tag. The DNA sequence encoding the RNA-binding zinc fingers of human TTP (TZF; UniProt: P26651, residues Ser102 to Ser169) was inserted between the NdeI and XhoI restriction sites of the pnYC plasmid[69] with a TEV-cleavable N-terminal MBP tag and a C-terminal StrepII tag. DNA constructs for the expression of the NOT9 module were previously described[70]. Subsequently, full-length human RESIST with an N-terminal TEV-cleavable His6–MBP tag and a C-terminal StrepII tag was expressed in Sf21 insect cells using the MultiBac baculovirus expression system[71,72] as previously described[73]. In brief, Sf21

cells were grown to a density of $2 \times 10^6$ cells per ml at 27 °C in Sf900II medium (Thermo Fisher Scientific), infected with the V1 generation His6–MBP–RESIST–StrepII baculovirus, and collected 48 h after they stopped dividing. The collected cells were resuspended in ice-cold protein buffer (50 mM HEPES pH 7.5, 150 mM NaCl, 5% (v/v) glycerol, 20 mM CHAPS, 25 mM imidazole) and lysed by sonication. The lysate was clarified by centrifugation at 40,000*g* for 40 min at 4 °C, filtered through a 0.45 µm nylon filter and loaded onto a 5 ml nickel-charged HisTrap column (Cytiva). Contaminants were removed by washing with lysis buffer supplemented with 40 mM imidazole, and His6–MBP–RESIST–StrepII was eluted in lysis buffer supplemented with 250 mM imidazole. The eluted protein was further purified by size-exclusion chromatography on the HiLoad Superdex 200 16/600 column (Cytiva) in buffer containing 50 mM HEPES pH 7.5, 150 mM NaCl, 5% (v/v) glycerol, 20 mM CHAPS. The peak fractions were pooled, concentrated with a centrifugal filter, flash-frozen in liquid nitrogen and stored at −80 °C. The RNA-binding zinc fingers of TTP (TZF) were expressed in *E. coli* BL21(DE3) Star cells (Thermo Fisher Scientific) in autoinduction medium[74] at 20 °C overnight as a fusion protein carrying an N-terminal, TEV-cleavable MBP tag and a C-terminal StrepII tag. Collected cells were resuspended in protein buffer (50 mM HEPES pH 7.5, 300 mM NaCl, 10% (w/v) sucrose) and lysed by sonication. The lysates were clarified by centrifugation at 40,000*g* for 40 min and loaded onto the 1 ml StrepTrap XT column (Cytiva). Contaminants were removed by washing with high-salt buffer (50 mM HEPES pH 7.5, 1 M NaCl, 10% (w/v) sucrose) before elution with lysis buffer supplemented with 50 mM biotin. Eluted protein was further purified by size-exclusion chromatography on the Superdex 200 26/600 column (Cytiva) in protein buffer supplemented with 2 mM DTT. The peak fractions were then pooled, concentrated with a centrifugal filter, flash-frozen in liquid nitrogen and stored at −80 °C. Finally, The NOT9 module was prepared as previously described[48].

## StrepTactin pull-down assay

StrepII-tagged MBP, as well as StrepII-tagged and SUMO-tagged SMARCA3 (residues 26–39) were produced in *E. coli* BL21 (DE3) Star cells (Thermo Fisher Scientific) grown in autoinduction medium overnight at 37 °C. Cells were resuspended in lysis buffer (50 mM HEPES, 500 mM NaCl, pH 7.5) and lysed using the Branson Ultrasonics Sonifier SFX550, the lysate was then cleared by centrifugation at 40,000*g* for 1 h at 4 °C. StrepII-tagged proteins (ANXA2R, MBP or SMARCA3) were incubated with StrepTactin Sepharose resin (Cytiva, 28935599). After incubation for 1 h, the beads were washed twice with 50 mM HEPES, 500 mM NaCl, pH 7.5, 0.03% Tween-20, once with 50 mM HEPES, 500 mM NaCl, pH 7.5 and once with binding buffer (50 mM HEPES, 200 mM NaCl, pH 7.5). Purified ANXA2(S100A) was added to the bead-bound proteins. After incubation for 1 h, the beads were washed four times with binding buffer and proteins were eluted with 50 mM biotin in binding buffer. The eluted proteins were analysed by SDS–PAGE followed by Coomassie blue staining.

For pull-downs of full-length human RESIST with CCR4–NOT subunits, purified His6–MBP–RESIST–StrepII or His6–MBP–StrepII were immobilized as bait through the C-terminal StrepII tag on streptavidin agarose resin prepared in-house. Then, 250 pmol of bait protein was incubated for 1 h in pull-down buffer (50 mM HEPES pH 7.5, 200 mM NaCl, 0.03% (v/v) Tween-20) at 6 °C under constant agitation. Unbound protein was removed after two washes with pull-down buffer, and 500 pmol of NOT9 module was incubated for 1 h with the bead-bound protein. Finally, the beads were washed three times with a pull-down buffer, and the bound proteins were eluted using a pull-down buffer supplemented with 50 mM biotin. The eluted proteins were analysed using SDS–PAGE followed by Coomassie blue staining.

## BlaER1 transduction and stimulation

Lentivirus was generated from HEK293T cells transfected with pLIP constructs (mCherry or human RESIST, either untagged or with

N/C-terminal HA tags) as described for BMM transduction above. Virus was overlaid onto BlaER1 cells, which were subsequently puromycin selected and differentiated as described previously[29]. After 5 days of differentiation, BlaER1 cells were stimulated with 2.5 µg ml$^{-1}$ doxycycline and fresh cytokines overnight. Medium with new cytokines, fresh doxycycline and ADU-S100 at a final concentration of 5 µg ml$^{-1}$ (Aduro) was added the next day. At the indicated timepoints, adherent and non-adherent cells were collected and lysed in TRK lysis buffer (Omega Biotek Total RNA kit) for RNA isolation. RNA was isolated as quickly as possible from lysates as described above, and converted to cDNA with Superscript reagents and RNase inhibitors as described above.

## Co-IP analysis

For IP analysis of RESIST from BMMs, BMMs were transduced with RESIST constructs or mCherry and treated with doxycycline, followed by DMXAA treatment, as described above. Cells ($0.6 × 10^6$–$2.4 × 10^6$) were collected at the timepoints indicated in figure legends, and lysed on ice in 300–600 µl lysis buffer (50 mM Tris-HCl pH 7.5, 0.2% NP-40, 5% glycerol, 100 mM NaCl, 1.5 mM MgCl$_2$, 1× protease inhibitor cocktail) for 30 min. The lysates were clarified by centrifugation at 18,213$g$ for 30 min at 4 °C and quantified using the bicinchoninic acid assay to ensure approximately equal amounts of input protein. One tenth of clarified lysate was diluted in Laemmli buffer for input sample. Supernatant was incubated with anti-HA magnetic beads (Thermo Fisher Scientific, 88836) with rotation for 3 h at 4 °C. Beads with lysates were then washed three times with lysis buffer (1 ml per wash), then immunoprecipitated proteins were eluted in 30–50 ml Laemmli buffer by boiling for 5 min. The samples were then analysed by immunoblotting.

For IP analysis of Flag–TTP from BMMs[75], N-terminally Flag-tagged mouse TTP constructs were cloned into pLIP as described above. Mouse Flag-TTP was transfected into semi-confluent 10 cm plates of HEK293T cells with N-terminally HA-tagged mouse RESIST or an equivalent amount of mCherry using Lipofectamine 2000 according to the manufacturer's instructions. 600 ng of TTP construct was co-transfected with 6 µg mCherry, and 660 ng of TTP construct was co-transfected with 6 µg RESIST. After transfection, cells were treated with doxycycline to induce expression for 24 h, then collected in PBS. Cells were lysed in hypotonic lysis buffer consisting of 10 mM Tris-HCl pH 7.5, 10 mM NaCl, 2 mM EDTA, 0.5% Triton X-100, and protease inhibitors for 10 min on ice with vortexing. NaCl was then adjusted to 150 mM and the samples were treated with 30 µg of RNase A on ice for 20 min with vortexing. The samples were clarified by centrifugation at 18,000$g$ for 30 min at 4 °C, and 10% of the sample was taken as the input sample and diluted in Laemmli buffer. The sample was incubated with 50 µl of Sigma anti-flag M2 agarose beads (Sigma Aldrich, M8823) equilibrated in lysis buffer with adjusted NaCl and resuspended in a total volume of 200 µl lysis buffer with adjusted NaCl per reaction. The samples were incubated with beads for 2 h with rotation at 4 °C, then washed five times with 1 ml wash buffer (50 mM Tris HCl pH 7.5, 300 mM NaCl, 0.05% Triton X-100, protease inhibitors). The remaining liquid was completely aspirated from the beads with a small-bore needle, and the beads were resuspended in 2× Laemmli buffer diluted in wash buffer before analysis by immunoblot.

## Immunoblot

Samples diluted in Laemmli buffer were run on 4–12% Bis-Tris protein gels (Invitrogen) and then transferred to PVDF membranes at 35 V for 90 min. Membranes were blocked in either Odyssey Licor PBS blocking buffer or in 2–5% non-fat dry milk diluted in TBST. The membranes were then probed with antibodies at 4 °C overnight diluted in 5% BSA TBST. The antibodies used in this study were as follows: rat anti-HA (Roche, 3F10, 1186742300, 1:1,000), mouse anti-actin (Santa Cruz Biotechnology, sc-47778, 1:1,000), rabbit anti-CNOT1 (Cell Signaling Technologies, 44613S, 1:1,000), rabbit anti-CNOT9 (Proteintech, 22503-1-AP, 1:500), rabbit anti-TTP (Millipore Sigma, ABE285, 1:1,000), rabbit anti-CNOT11 (Sigma-Aldrich, HPA069823, 0.4 µg ml$^{-1}$), rabbit anti-ZFP36L1 (Cell Signaling Technologies, 30894S, 1:1,000), rabbit anti-ZFP36L2 (Abcam, ab70775, 1:1,000), and rabbit anti-SP140 (Covance, as previously described[4]; 1:1,000) and rabbit anti-Flag for Flag–TTP IPs (Thermo Fisher Scientific, PA1-984B, 1:1,000).

## In vivo *L. pneumophila* infections

*L. pneumophila* infections were performed as described previously[4]. In brief, JR32Δ*flaA L. pneumophila* (from the laboratory of D. Zamboni) was streaked from a frozen glycerol stock onto BCYE plates. A single colony was used to streak an approximately 4 cm$^2$ patch that was subsequently grown for 2–3 days. Bacteria were diluted in water and the optical density was measured at 600 nm to determine the bacterial concentration. Bacteria were then diluted to a final concentration of $2.5 × 10^6$ bacteria per ml in sterile PBS. Mice were anaesthetized with a mixture of xylazine and ketamine through intraperitoneal injection, then 40 µl of diluted bacteria was administered intranasally (final infectious dose of $10^5$ bacteria per mouse). At 96 h after infection, mice were euthanized and the lungs were homogenized in 5 ml of autoclaved MilliQ water. Lung homogenate was diluted and plated on BCYE plates, and colony-forming units were enumerated after 4 days of growth. One infection of *Sp140$^{-/-}$ Resist1/2$^{+/+}$* and *Sp140$^{-/-}$ Resist1/2$^{-/-}$* littermates was performed, which reproduced the results obtained with non-littermate infections.

## AlphaFold structure predictions

AlphaFold-Multimer (v.2.3.2)[76] was run on equipment hosted by the Cal Cryo EM facility comprising an Nvidia GPU and >72 TB of storage space. Mouse RESIST and CCR4–NOT amino acid sequences were from NCBI (RESIST, XP_006517870.1; CNOT1 M-HEAT, XP_036009857.1, with a start codon followed by residues 815–1007; CNOT11, NP_082319.1; CNOT1 N-MIF4G, XP_036009857.1, residues 1–695; CNOT10, NP_705813.2; CNOT9, NP_067358.1). In brief, AlphaFold was run in multimer mode on RESIST with CNOT1 M-HEAT, CNOT9 or CNOT1 N-MIF4G/CNOT10/CNOT11 with the default settings, the max template date specified as 2023-01-01 and --db_preset=full_dbs. Output models were visualized in ChimeraX (v.1.6.1), and aligned using the matchmaker command. PAE plots and structures coloured by pLDDT were visualized with PAEViewer (v.1.0)[77].

## Gene disruption in BMMs with Cas9–gRNA electroporation

BMMs were collected in PBS and electroporated with Cas9 2 NLS nuclease (Synthego) complexed with gRNAs (Synthego, sgRNA EZ kits) and Alt-R Cas9 Electroporation Enhancer (IDT, 1075916) in Lonza P3 buffer (Lonza, V4XP-3032) with buffer supplement as described previously[78]. Electroporation was performed using the Lonza 4D-Nucleofector Core Unit (AAF-1002B) using the program CM-137. Electroporated BMMs were immediately plated in BMM medium. A half-medium exchange was performed every 2 days until day 10 after bone marrow collection, when BMMs were seeded for downstream assays. The knockout efficiency was evaluated by immunoblotting or PCR analysis of genomic DNA for targeted regions followed by ICE analysis (Synthego) as indicated in figure legends. *Gm21188* (*Resist1*) was genotyped with Primestar PCR reagents and the primers F1 (ATTGAGAAATCCGTTTGTAATGGG) and R1 (TAGGCGAATTTCGTGGCACA) according to the manufacturer's instructions with an annealing temperature of 55 °C, or using Q5 PCR reagents and primers F2 (TTGAGAAATCCGTTTGTAATGGG) and R2 (GCCTTTCTCCGGATTCACGA) with the cycling conditions: 98 °C for 3 min; 35 cycles of 98 °C for 10 s, 63.5 °C for 20 s and 72 °C for 1 min 5 s. *Gm36079* (*Resist2*) was genotyped with F (TGGTATTCTCTAGAGAT AACATCACAGCACCTACTTACTCC) and R (CCTCCCCTCGCCATCACTG CCTG) using the Phusion GC PCR reagents and cycling conditions of: 98 °C for 30 s, then 30–35 cycles of 98 °C for 10 s, 72 °C for 15 s, 72 °C for 1 min. Disruption of *Rc3h1* was determined by PCR (F, CACACTATG

TGCTGACTGTATCTACAGAAG; R, TCCCCTCAGGTAAAACAGTGC; cycling: 98 °C for 30 s, then 30 cycles of 98 °C for 10 s, 60 °C for 5 s and 72 °C for 1 min) with Phusion GC PCR reagents. Disruption of *Rc3h2* was determined by PCR (Q5 PCR reagents with F2, AGGGCATAAGATG TTGCACAGA; R2, ACTGCTAACCCGAGCATCAG; and cycling of 98 °C for 3 min, then 35 cycles of 98 °C for 10 s, 60 °C for 20 s and 72 °C for 40 s). PCRs were cleaned by gel extraction, Ampure XP beads (Beckman Coulter) or treatment with FAST-AP and ExoI before submission for Sanger sequencing. ICE analysis was performed with the Synthego ICE online tool (https://ice.editco.bio/#/). The gRNA sequences used in this study were as follows: *Gm21188/Gm36079* gRNA 1, GCUGGGCCUCUUGCACC AGA; *Gm21188/Gm36079* gRNA 2, CGACGAUGGCGGUGACUACC; *Cnot1* gRNA 1, UGUGAAUCGGCACGGUCCUG; *Cnot1* gRNA 2, ACUC AUUCAGGAUUAACAGA; *Cnot11* gRNA 1, UCCAUCAAGGCAAUCUGG CG; *Cnot11* gRNA 2, GCUGAGCAUCAUCUCGGAGG; *Cnot9* gRNA 1, CAUU GCAAACUCUGUUAGAC; *Cnot9* gRNA 2, GCCUACUGCACUAGCCC AAG; *Zfp36* gRNA 1, CAUGACCUGUCAUCCGACCA; *Zfp36* gRNA 2, CUUCAUCCACAACCCCACCG; *Zfp36l1* gRNA 1, AAAAAUGGUGGCGG ACACGA; *Zfp36l1* gRNA 2, ACGGGCAAAAGCCGAUGGUG; *Zfp36l2* gRNA 1, CAAGAAGUCGAUAUCGUAGA; *Zfp36l2* gRNA 2, GAGAGCGGCACGUGC AAGUA; *Rc3h1* gRNA, CAAAUGGGCAAGCCUUACGG; *Rc3h2* gRNA, UCGGUGAAGUUUAUUCAAGC.

## TTP EMSA

The substrate RNAs were generated by in vitro transcription (IVT). For the *IFNB1* WT RNA substrate, the ARE in the 3′-UTR of the *IFNB1* mRNA (GenBank: NM_002176.4; nucleotides 740–825) was synthesized as a gene fragment (Azenta) with an upstream T7 promoter and 17 random nucleotides downstream. All adenosine residues between nucleotides 758 and 825 were mutated to cytosine for the *IFNB1-MUT* RNA substrate. The gene fragments were amplified by PCR, and the purified PCR products were used as templates for IVT using the HiScribe T7 High Yield RNA Synthesis Kit (NEB). IVT products were separated by size-exclusion chromatography on the Superdex 200 increase 10/300 GL in buffer containing 10 mM HEPES pH 7.5, 200 mM NaCl. The fractions containing the intact RNA substrates were pooled, ethanol precipitated and resuspended in RNase-free water. Electrophoretic mobility shift assay (EMSA) binding reactions contained 50 nM substrate RNA and 50–800 nM TZF protein. The reactions were carried out for 15 min at 37 °C in a buffer containing 20 mM PIPES pH 6.8, 10 mM KCl, 40 mM NaCl, 2 mM Mg(OAc)$_2$, 3% (v/v) Ficoll 400 and 0.05% (v/v) NP-40. The RNA–protein complexes were analysed by electrophoresis on a nondenaturing polyacrylamide gel in 0.5× TBE buffer, pH 8.3, at 10 V cm$^{-1}$. Gels were stained in 0.5× TBE pH 7.5 with 1× SYBR Gold (Thermo Fisher Scientific) for 5 min before analysis. Images were quantified using FiJi[79].

## Viral infections of BMMs

Day 7 BMMs were generated as described above, and 250,000 cells were infected with viruses at the indicated multiplicities of infection in a non-tissue-culture-treated 12-well plate. For MCMV-GFP and MHV68-GFP, cells were infected for 3–4 h in serum-free RPMI supplemented with penicillin–streptomycin and glutamine in a low volume of inoculum. MCMV-GFP and MHV68-GFP were a gift from L. Coscoy and B. A. Glaunsinger. MHV68-GFP and MCMV-GFP titre was estimated by infection of 3T3 cells, and calculated by the assumption that a viral dilution resulting in 100% of infected 3T3 cells corresponds to an approximate multiplicity of infection of 5. For Sendai-GFP infections (ViraTree), cells were infected for 1.5 h in serum-free RPMI supplemented with penicillin–streptomycin and glutamine in a low inoculum volume. After infection, medium was replaced and BMMs were cultured for an additional 20–24 h before cells were collected in PBS, stained with Ghost Dye Far Red 780, fixed with the BD Cytofix Cytoperm kit according to the kit instructions, then washed and analysed by flow cytometry. Data were analysed using FlowJo.

## Immunofluorescence microscopy

BMMs were seeded on glass coverslips between $0.5 \times 10^6$ and $1 \times 10^6$ per slip in BMM medium lacking antibiotics and rested overnight. BMMs were stimulated with 100 µg ml$^{-1}$ DMXAA for 8 h, fixed in 4% freshly prepared paraformaldehyde (Electron Microscopy Sciences) for 10 min at room temperature, then permeabilized in freshly made 0.2% Triton X-100 and 0.2% BSA in PBS on ice for 10 min. The coverslips were washed three times with PBS and then blocked in goat serum and FC-block (TruStain FcX PLUS, anti-mouse CD16/32) for 1 h, then incubated overnight at 4 °C in primary antibody diluted in PBS with 1% Tween-20 and 1% BSA. Primary antibodies and dilutions were mouse anti-PML (Millipore Sigma, 05-718, 1:100), rat anti-HA (Roche, 11867423001, 1:200), rabbit anti-fibrillarin (Abcam, ab166630, 1:100). Coverslips were washed three times in PBS, then incubated in secondary antibody diluted 1:1,000 in PBS with 1% Tween-20 and 1% BSA for 2–3 h at room temperature. Secondary antibodies used were donkey anti-rat Alexa Fluor 488 (Invitrogen, A21208), goat anti-mouse Alexa Fluor 647 (Invitrogen, A21236), goat anti rabbit 647 (Life technologies, A21244). After three PBS washes of coverslips, nuclei were stained with DAPI at 1 µg ml$^{-1}$ in PBS for 10 min at room temperature, followed by three PBS washes. The coverslips were mounted in Vectashield mounting medium (Vector Laboratories, H-1000-10). Coverslip edges were then sealed with clear nail polish before imaging on a Zeiss LSM 880 NLO AxioExaminer at ×63 magnification.

## IF image processing and quantification

For independent experiments as indicated in figure legends, *z* stacks were processed by Imaris File Converter v.10.0.1 followed by Imaris Stitcher v.9.9.1. Images were Gaussian filtered (0.132 µm) and screenshots were generated for figures. Surfaces for DAPI, HA–SP140, PML and fibrillarin were generated in Imaris using split-touching of 1 µm for HA–SP140, PML and fibrillarin surfaces and 5 µm for DAPI surfaces. Surface statistics were then exported. Surfaces were first filtered as DAPI$^+$ (within 2 s.d. below average fluorescence intensity of DAPI surfaces) and then filtered by size (over 0.2 µm$^3$). HA–SP140 NBs were considered to overlap fibrillarin if the fibrillarin intensity mean for an HA–SP140 surface was within 2 s.d. of the average fluorescence intensity for fibrillarin surfaces, and vice versa for fibrillarin surfaces overlapping with HA–SP140 surfaces. The same criteria for overlap were applied to HA–SP140 and PML surfaces. The mean fluorescence intensities of surfaces were calculated for over 100 nuclei for each independent experiment.

## Statistics and reproducibility

All results except for HA–SP140 CUT&RUN and AlphaFold predictions were repeated at least twice in independent experiments. Statistics and graphs for all experiments except for RNA-seq, ATAC–seq and CUT&RUN experiments were generated using GraphPad Prism (v.10.0.2). For data with two groups of comparison, *P* values were calculated with two-tailed *t*-tests using Welch's correction or two-way ANOVA, as described in the figure legends. For data with more than two comparison groups, ANOVA was used. We found that the residuals for our RT–qPCR data were not normally distributed and for these data we therefore performed ANOVA on log$_{10}$-transformed data, which generated more normally distributed residuals based on Q–Q plots, therefore more appropriate for ANOVA tests. We used one-way Welch's and Brown–Forsythe ANOVA without assuming data sphericity or equal variance, for data with multiple genotypes and one treatment condition, with a more-conservative post hoc Dunnett's T3 multiple-comparison correction for log$_{10}$-transformed data and less-conservative post hoc FDR correction for multiple comparisons (*Q* = 0.001, two-stage linear step-up procedure of Benjamini, Krieger, and Yekutieli) for all other data. Two-way ANOVA was performed for data with more than two comparison groups and/or multiple timepoints of measurement, with a full model including an interaction term, as we found that the effect of genotype varied across time.

For two-way ANOVA, we did not assume sphericity, and used post hoc Tukey's multiple-comparison test or Šidák's multiple-comparison correction as described in the figure legends. For non-normally distributed data (*L. pneumophila* in vivo infections), Kruskal–Wallis one-way ANOVA with Dunn's multiple-comparison test was used. The mean for all data was graphed, and replicates are individually represented by dots. Error bars indicate the s.e.m. Data shown in each figure represent the provided source raw data; statistical test details are also provided in the Source data. Replicates in RT–qPCR, ELISA or viral infections represent separate wells within an experiment, while replicates in *L. pneumophila* infections represent individual mice from three combined experiments. Replicate numbers (*n*) are represented in the figures, legends and Source data. For all experiments, samples were grouped based on genotype and treatment group and were not further randomized. Investigators were not blinded to experimental groups, and statistical methods were not used to predetermine sample size.

## Schematics

All schematics were generated in BioRender by R.E.V. Schematics are available at https://BioRender.com/owrrzg2 for Fig. 4h, https://BioRender.com/0kcpnq3 and https://BioRender.com/rig1le4 for Fig. 5h, and https://BioRender.com/22n4jit for Extended Data Fig. 4a.

## Reporting summary

Further information on research design is available in the Nature Portfolio Reporting Summary linked to this article.

## Data availability

HA–SP140 anti-HA CUT&RUN data are available at the Gene Expression Omnibus (GEO: GSE269315). RNA-seq data for *Ifnar*[−/−] and *Sp140*[−/−]*Ifnar*[−/−] BMMs are available under GEO accession GSE269761. RNA-seq and ATAC–seq data for B6 and *Sp140*[−/−] BMMs are available under GEO accessions GSE269808 and GSE269811, respectively. The mm10 genome is available online (https://genome.ucsc.edu/cgi-bin/hgGateway?db=mm10). Source data are provided with this paper.

## Code availability

Source code for all analysis scripts and pipelines is available at GitHub (https://github.com/adziulko/The-SP140-RESIST-pathway).

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

**Acknowledgements** We thank the members of the Vance and Barton laboratories for input and advice; E. Brydon for technical assistance; J. Li (UC Davis Bioinformatics) for preliminary analysis of ATAC–seq and CUT&RUN datasets; K. Heydari, M. Delcroix, H. Dhaliwal (UC Berkeley Cancer Research Lab) for technical assistance and E. Witt for advice on statistical and bioinformatic analyses. R.E.V. is an Investigator of the Howard Hughes Medical Institute, and research in his laboratory is supported by NIH grants AI075039, AI063302 and AI155634, a Bakar Spark award, and an Emerging Pathogens Initiative grant from HHMI. K.C.W. was supported by the National Science Foundation Graduate Research Fellowship Program. F.P., D.J.T. and E.V. were supported by the Intramural Research Program of the National Institutes of Health. F.P. was further supported by a Walter Benjamin postdoctoral fellowship from the German Research Foundation (Deutsche Forschungsgemeinschaft, project number, 531520533). S.A.F. was supported by a Postdoctoral Fellowship from EMBO (ALTF 617-2021) and the Swiss National Science Foundation (P500PB_206801).

**Author contributions** Conceptualization: K.C.W. and R.E.V. Data analysis: K.C.W., A.D., J.A., P.A., A.X.C. and E.B.C. Investigation: K.C.W., A.D., J.A., F.P., A.X.C., G.Y.L., O.V.L., D.J.T., M.M.G., R.C., S.A.F., P.A., D.I.K., L.C., E.V., E.B.C. and R.E.V. Writing: K.C.W., A.D. and R.E.V. Resources: A.L., H.D., A.Y.L., B.A.G., L.C. and G.Y.L. Funding acquisition: K.C.W. and R.E.V.

**Competing interests** R.E.V. is on the scientific advisory boards of Tempest Therapeutics and X-biotix.

**Additional information**
**Correspondence and requests for materials** should be addressed to Russell E. Vance.

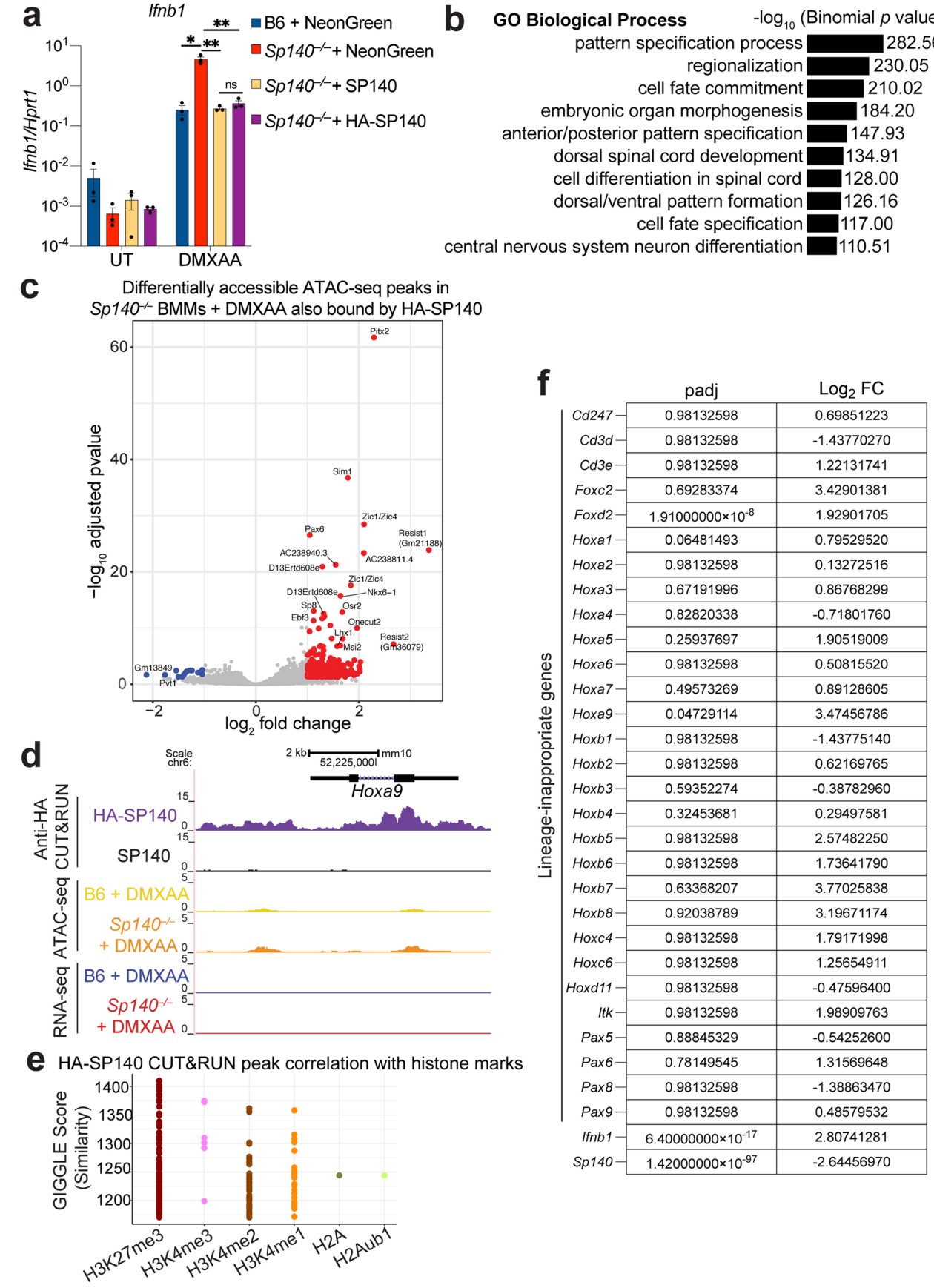

**a** *Ifnb1*

B6 + NeonGreen
*Sp140*⁻/⁻ + NeonGreen
*Sp140*⁻/⁻ + SP140
*Sp140*⁻/⁻ + HA-SP140

**b** GO Biological Process                    -log₁₀ (Binomial *p* value)

| pattern specification process | 282.56 |
| regionalization | 230.05 |
| cell fate commitment | 210.02 |
| embryonic organ morphogenesis | 184.20 |
| anterior/posterior pattern specification | 147.93 |
| dorsal spinal cord development | 134.91 |
| cell differentiation in spinal cord | 128.00 |
| dorsal/ventral pattern formation | 126.16 |
| cell fate specification | 117.00 |
| central nervous system neuron differentiation | 110.51 |

**c** Differentially accessible ATAC-seq peaks in
*Sp140*⁻/⁻ BMMs + DMXAA also bound by HA-SP140

**d**

**e** HA-SP140 CUT&RUN peak correlation with histone marks

**f**

| | padj | Log₂ FC |
|---|---|---|
| *Cd247* | 0.98132598 | 0.69851223 |
| *Cd3d* | 0.98132598 | -1.43770270 |
| *Cd3e* | 0.98132598 | 1.22131741 |
| *Foxc2* | 0.69283374 | 3.42901381 |
| *Foxd2* | 1.91000000×10⁻⁸ | 1.92901705 |
| *Hoxa1* | 0.06481493 | 0.79529520 |
| *Hoxa2* | 0.98132598 | 0.13272516 |
| *Hoxa3* | 0.67191996 | 0.86768299 |
| *Hoxa4* | 0.82820338 | -0.71801760 |
| *Hoxa5* | 0.25937697 | 1.90519009 |
| *Hoxa6* | 0.98132598 | 0.50815520 |
| *Hoxa7* | 0.49573269 | 0.89128605 |
| *Hoxa9* | 0.04729114 | 3.47456786 |
| *Hoxb1* | 0.98132598 | -1.43775140 |
| *Hoxb2* | 0.98132598 | 0.62169765 |
| *Hoxb3* | 0.59352274 | -0.38782960 |
| *Hoxb4* | 0.32453681 | 0.29497581 |
| *Hoxb5* | 0.98132598 | 2.57482250 |
| *Hoxb6* | 0.98132598 | 1.73641790 |
| *Hoxb7* | 0.63368207 | 3.77025838 |
| *Hoxb8* | 0.92038789 | 3.19671174 |
| *Hoxc4* | 0.98132598 | 1.79171998 |
| *Hoxc6* | 0.98132598 | 1.25654911 |
| *Hoxd11* | 0.98132598 | -0.47596400 |
| *Itk* | 0.98132598 | 1.98909763 |
| *Pax5* | 0.88845329 | -0.54252600 |
| *Pax6* | 0.78149545 | 1.31569648 |
| *Pax8* | 0.98132598 | -1.38863470 |
| *Pax9* | 0.98132598 | 0.48579532 |
| *Ifnb1* | 6.40000000×10⁻¹⁷ | 2.80741281 |
| *Sp140* | 1.42000000×10⁻⁹⁷ | -2.64456970 |

Lineage-inappropriate genes

**Extended Data Fig. 1** | See next page for caption.

**Extended Data Fig. 1 | SP140 predominantly represses chromatin accessibility and binds genes involved in development, although these genes are not differentially expressed in the absence of SP140.** a. RT-qPCR for transduced *Sp140⁻/⁻* BMMs, 8 h 100 μg/mL DMXAA, for CUT&RUN samples. Mean +/− s.e.m are plotted, n = 3 wells of cells, * = $p < 0.05$, ** = $p < 0.005$, *** = $p < 0.0005$, **** = $p < 0.0001$, ns = not significant, one-way ANOVA with Dunnett's T3 post-hoc correction. $p = 0.0092$ for B6 vs. *Sp140⁻/⁻* + NeonGreen, 0.0017 for *Sp140⁻/⁻* + NeonGreen vs. *Sp140⁻/⁻* + SP140, 0.0013 for *Sp140⁻/⁻* + NeonGreen vs. *Sp140⁻/⁻* + HA-SP140. Exact *p* values are in Source Data. Results representative of four independent experiments. b. Top 10 GO terms for HA-SP140 bound genes in anti-HA CUT&RUN. Adjusted *p* values calculated with two-sided binomial test and multiple comparison corrections (GREAT)[67]. c. Volcano plot of differentially accessible ATAC-seq peaks in DMXAA-treated *Sp140⁻/⁻* vs. B6 BMMs, filtered by genes also bound by HA-SP140 in anti-HA CUT&RUN. Blue dots indicate genes with $\log_2$ fold change < −1 and adjusted *p* (padj) <0.05; red dots indicate genes with $\log_2$ fold change > 1 and padj <0.05. for differential ATAC-seq peak accessibility in DMXAA-treated *Sp140⁻/⁻* BMMs. Adjusted *p*-values are in Source Data and were calculated with DeSeq2 (Fig. 2a). d. Alignment of reads from HA-SP140 or untagged SP140 anti-HA CUT&RUN, and RNA-seq/ATAC-seq of *Sp140⁻/⁻* and B6 BMMs, at *Hoxa9*. e. GIGGLE similarity score for HA-SP140 CUT&RUN peak overlap with publicly available ChIP-seq datasets for indicated histone marks[66]. f. Table of adjusted *p* (padj) and $\log_2$ fold change for "lineage-inappropriate SP140-regulated" genes[15] from DMXAA-treated *Sp140⁻/⁻* and B6 BMMs RNA-seq. Adjusted *p*-values calculated as in c.

**a**

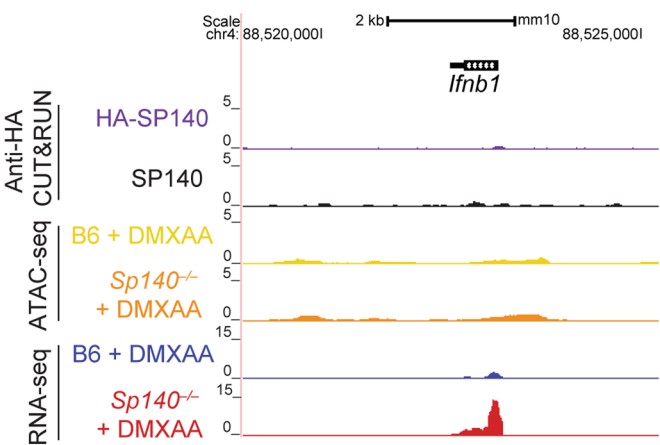

**b**

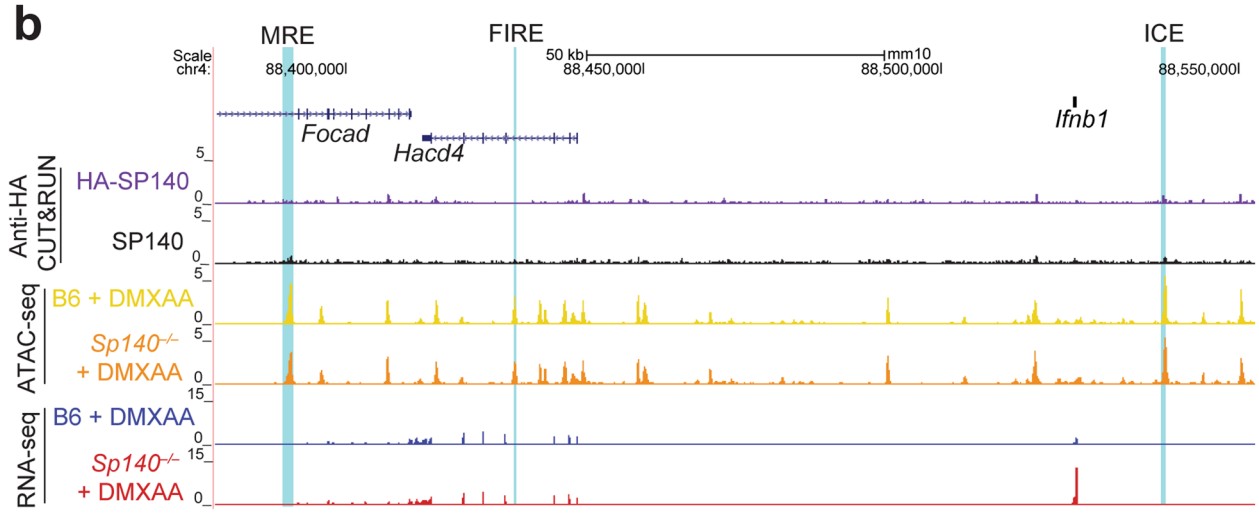

**Extended Data Fig. 2 | SP140 does not bind the *Ifnb1* locus or known regulatory elements.** a. Alignment of reads at *Ifnb1* from HA-SP140 or untagged SP140 anti-HA CUT&RUN, ATAC-seq of DMXAA-treated *Sp140*[-/-] and B6 BMMs, and RNA-seq of DMXAA-treated *Sp140*[-/-] and B6 BMMs. b. Alignment of reads from HA-SP140 or untagged SP140 anti-HA CUT&RUN, and ATAC-seq/RNA-seq of *Sp140*[-/-] and B6 BMMs treated with DMXAA, at the *Ifnb1* regulatory elements ICE[31,32], FIRE[30], and the MRE[29].

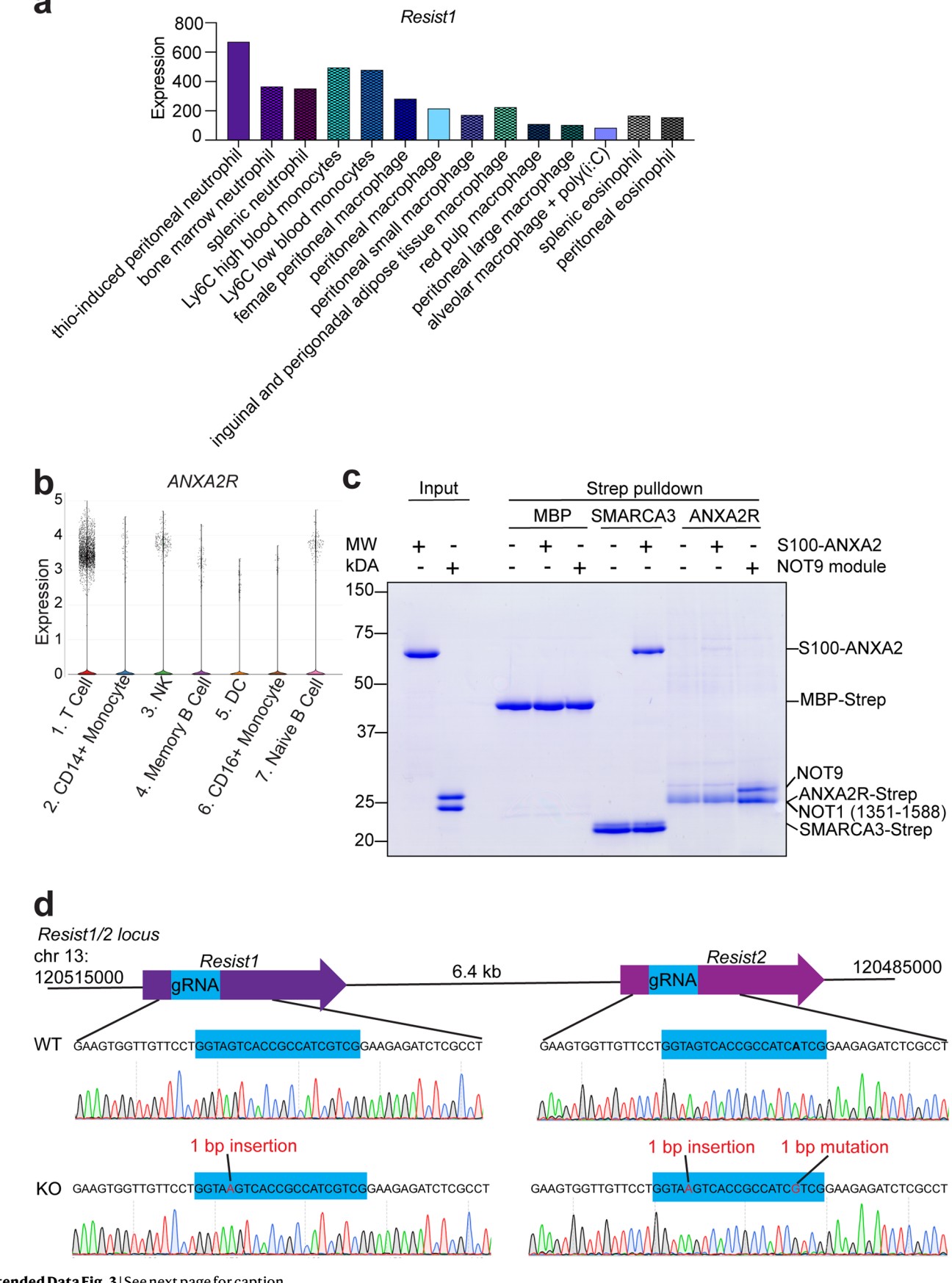

**Extended Data Fig. 3** | See next page for caption.

**Extended Data Fig. 3 | ANXA2R/RESIST expression in humans and mice, assessment of binding to Annexin 2, and generation of *Gm21188*/*Resist1*<sup>−/−</sup> *Gm36079/Resist2*<sup>−/−</sup> mice.** a. Expression values (DESeq2 normalized values) for *Gm21188* (*Resist1*) for all cell types with high expression (>80 for expression value). Data from immgen.org. b. Expression of *ANXA2R* in human PBMC single-cell RNAseq data from Immune Cell Atlas (data from https://singlecell.broadinstitute.org/single_cell/study/SCP345/ica-blood-mononuclear-cells-2-donors-2-sites?scpbr=immune-cell-atlas#study-summary) c. Pull-down assay of recombinant STREP-ANXA2R and STREP-SMARCA3 (residues 26–39) upon incubation with ANXA2-S100A. Results representative of two independent experiments. For gel source data, see Supplementary Fig. 1. d. Schematic of *Resist1/2 (Gm21188/Gm36079)* locus with protein coding sequences indicated with purple arrows and gRNA targeted sequence indicated in blue. *Resist2* contains a SNP within the gRNA targeting sequence (in bold). WT traces are indicated at guide-targeted regions for both *Resist1* and *Resist2*. Sequence traces from *Sp140*<sup>−/−</sup>*Resist1*<sup>−/−</sup>*Resist2*<sup>−/−</sup> mice (KO) are below with indicated mutations in *Resist1/2*.

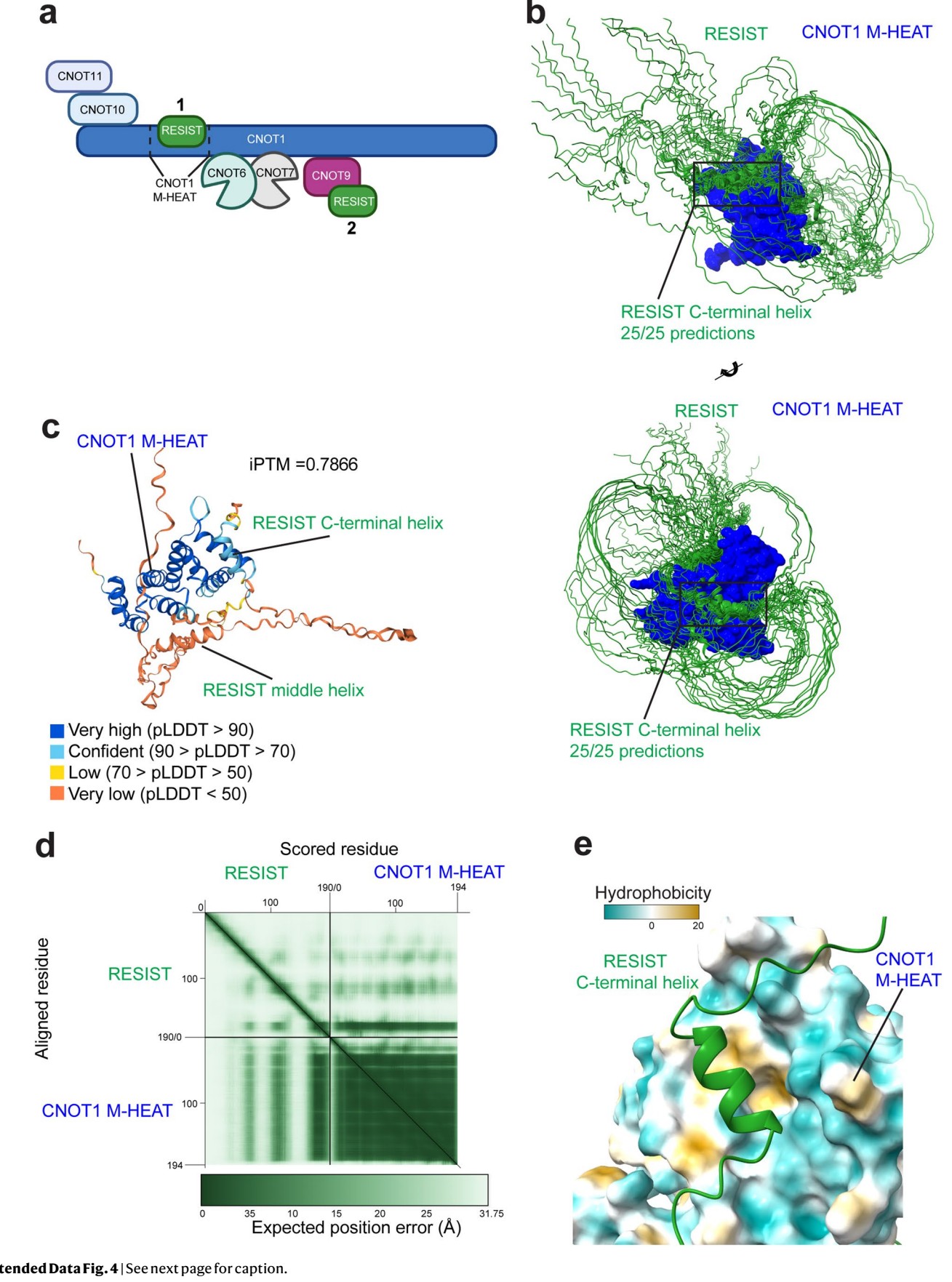

**Extended Data Fig. 4** | See next page for caption.

**Extended Data Fig. 4 | AlphaFold predictions suggest RESIST likely binds the CNOT1 M-HEAT domain.** a. Schematic of CCR4-NOT subunit and RESIST complexes produced by Alphafold Multimer. Schematic was generated in BioRender.com. b. Aligned AlphaFold models of RESIST with the CNOT1 M-HEAT domain. c. The top-scoring structural prediction of the RESIST and CNOT1 M-HEAT complex, coloured by pLDDT. d. AlphaFold PAE plot of predicted RESIST and CNOT1 M-HEAT complex. Plot generated with PAEViewer[77]. e. Depiction of RESIST binding to a hydrophobic patch on the CNOT-1 M-HEAT domain (CNOT1 coloured by hydrophobicity).

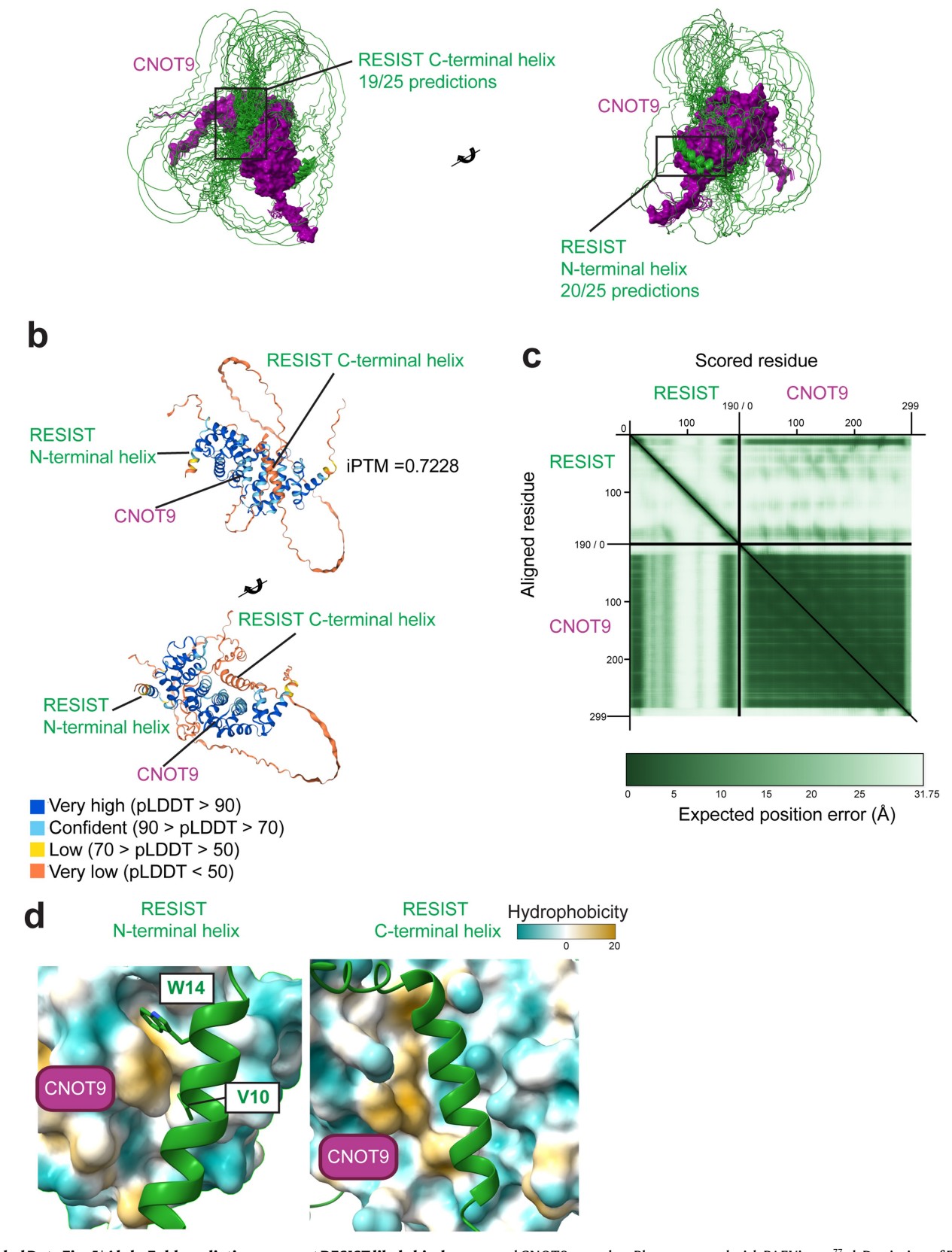

**Extended Data Fig. 5 | AlphaFold predictions suggest RESIST likely binds the CNOT9 subunit.** a. Aligned structural predictions of RESIST interactions with CNOT9. b. Second-ranked Alphafold structural prediction of RESIST and CNOT9 complex coloured by pLDDT. c. AlphaFold PAE plot of predicted RESIST and CNOT9 complex. Plot generated with PAEViewer[27]. d. Depiction of RESIST binding to multiple hydrophobic patches on CNOT9 (CNOT9 coloured by hydrophobicity).

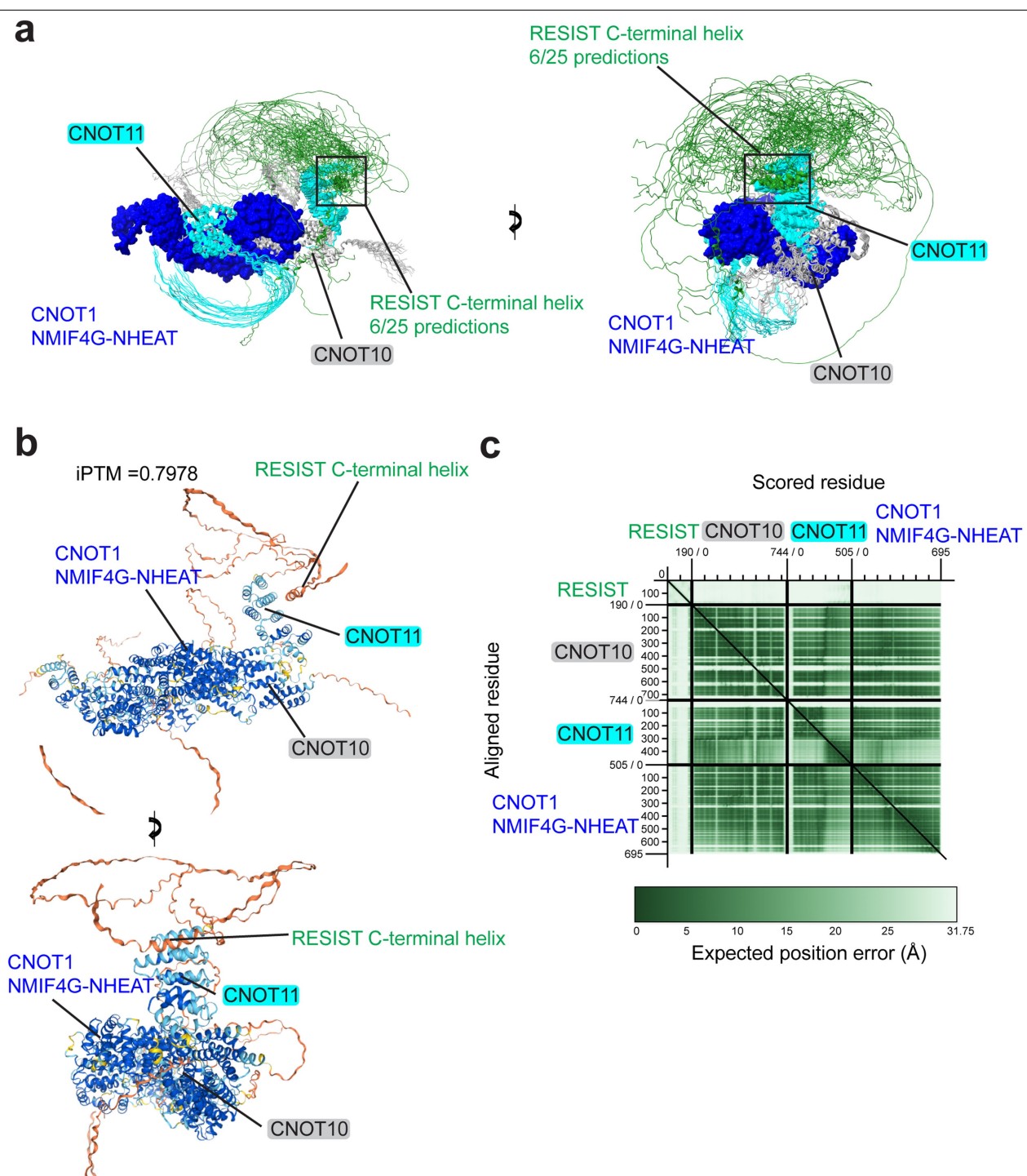

**Extended Data Fig. 6 | AlphaFold predictions suggest RESIST likely does not bind CNOT11.** a. Aligned AlphaFold structure predictions of RESIST with the CNOT1 NMIF4G-NHEAT domains, CNOT10, and CNOT11. b. Highest-scoring AlphaFold structure prediction of RESIST and CNOT1/CNOT10/CNOT11 complex coloured by pLDDT. c. AlphaFold PAE plot of predicted RESIST and CNOT1/CNOT10/CNOT11 complex. Plot generated with PAEViewer[77].

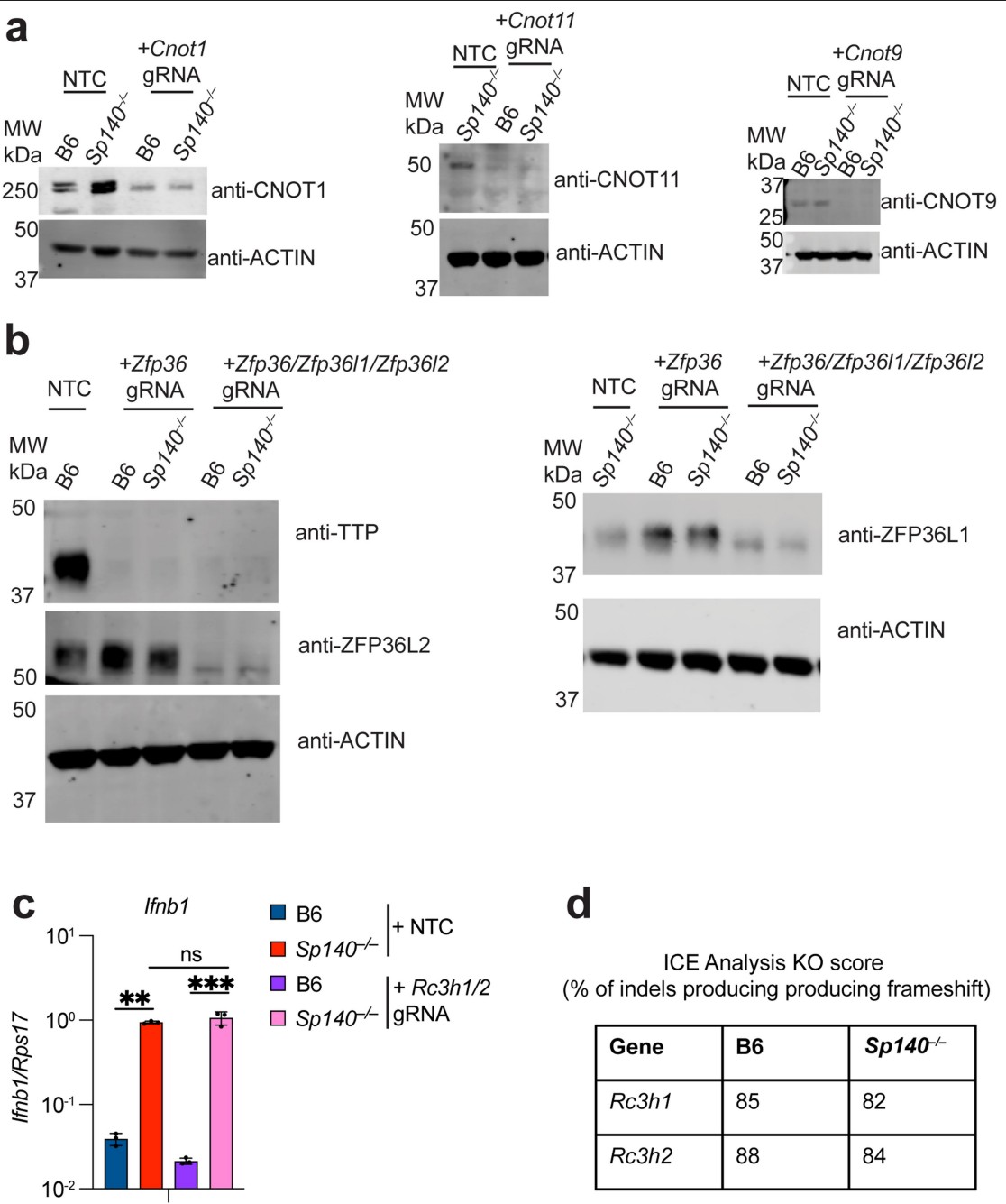

**Extended Data Fig. 7 | Assessment of CCR4-NOT subunit, TTP family, and ROQUIN1/2 knockout efficiency and role of ROQUIN1/2 in *Ifnb1* regulation in BMMs.** a. Immunoblot of B6 or *Sp140*−/− BMMs electroporated with indicated gRNAs for indicated CCR4-NOT subunits for experiments shown in Fig. 4b. Actin blots represents loading controls. For gel source data, see Supplementary Fig. 1. b. Immunoblot of B6 or *Sp140*−/− BMMs electroporated with indicated gRNAs for indicated TTP family members for experiments shown in Fig. 4f after 8 h of 100 μg/mL DMXAA. Actin blots represents loading controls. Results representative of two independent experiments. For gel source data, see Supplementary Fig. 1. c. RT-qPCR for *Ifnb1* in BMMs electroporated with NTC

gRNAs or gRNAS targeting *Rc3h1*/*Rc3h2* (genes encoding ROQUIN1/2) after 8 h of 100 μg/mL DMXAA treatment. Data mean +/− s.e.m are plotted, n = 3 wells of cells, * = p < 0.05, ** = p < 0.005, *** = p < 0.0005, **** = p < 0.0001, ns = not significant, one-way ANOVA with Dunnett's T3 post-hoc correction. Adjusted p = 0.0025 for B6 + NTC vs. *Sp140*−/− + NTC, 0.8028 for *Sp140*−/− + NTC vs. *Sp140*−/− + *Rc3h1*/*2* gRNA, and 0.0002 for B6 + *Rc3h1*/*2* gRNA vs. *Sp140*−/− + *Rc3h1*/*2* gRNA. Additional exact adjusted p values and statistical test results are provided in Source Data. Results representative of two independent experiments. d. Knockout efficiency for *Rc3h1*/*Rc3h2* in BMMs electroporated with gRNAs targeting *Rc3h1* and *Rc3h2* for results shown in c.

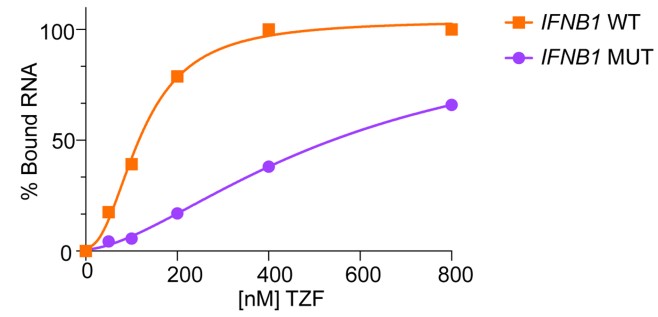

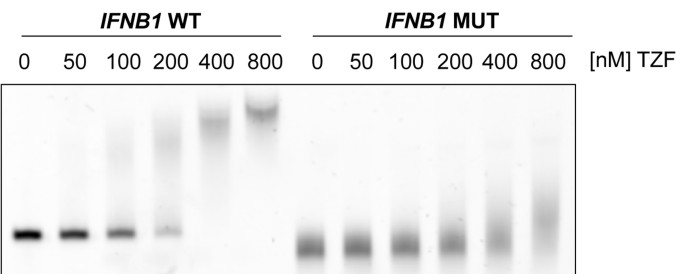

**Extended Data Fig. 8 | The TTP zinc finger domain (TZF) binds the *Ifnb1* 3'UTR in an ARE-dependent manner.** Top: binding curve of TZF to SYBR-Gold labelled *Ifnb1* 3'UTR RNA for either WT (orange) or ARE mutant (purple), quantified from bottom. Bottom: representative electrophoretic mobility shift assay (EMSA) of TZF to ARE-WT or mutant *Ifnb1* 3'UTR. For gel source data, see Supplementary Fig. 1. Results representative of two independent experiments.

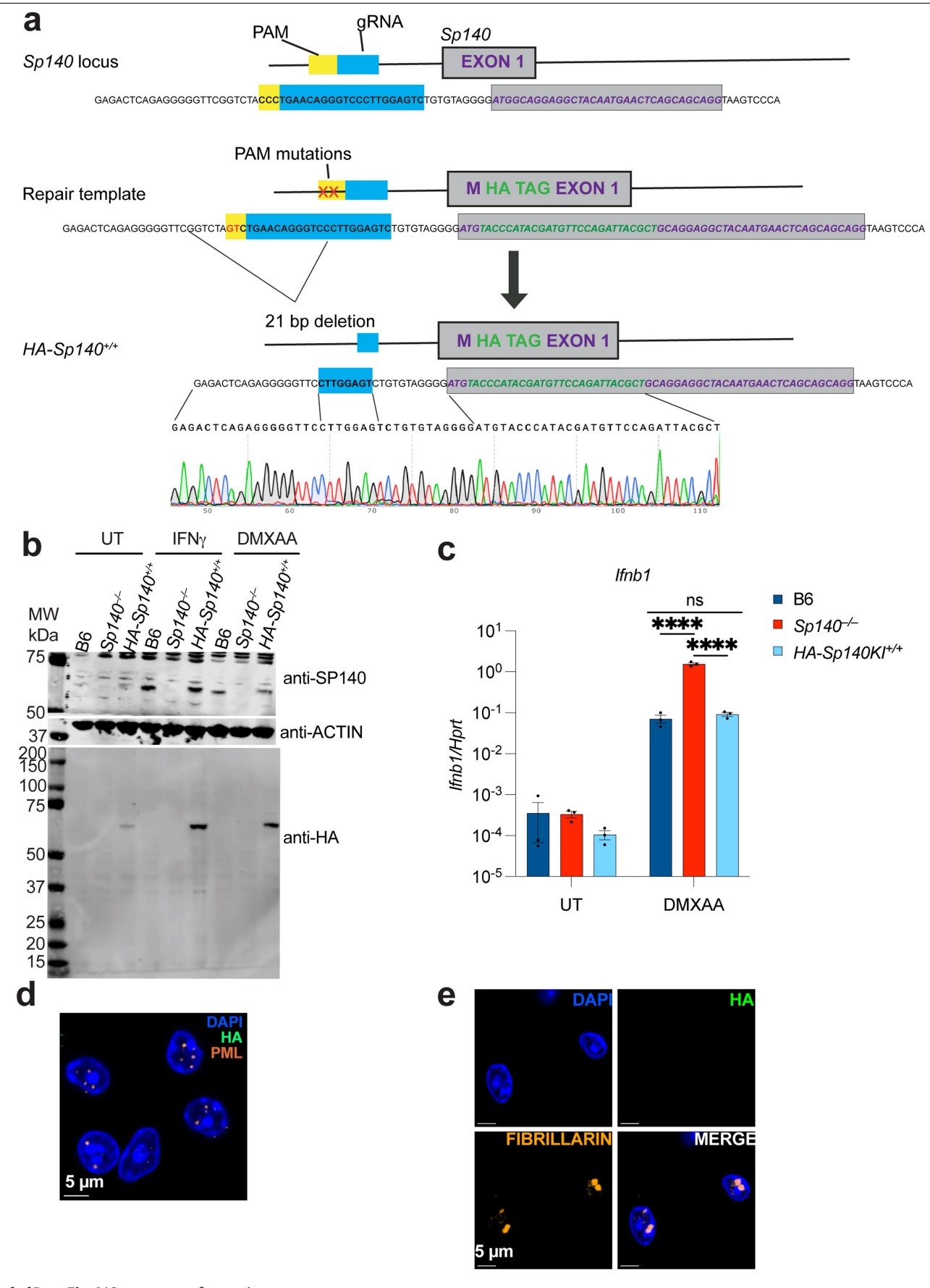

**Extended Data Fig. 9** | See next page for caption.

**Extended Data Fig. 9 | Generation and validation of *HA-Sp140* knock-in mice.** a. Schematic of gene-targeting strategy to generate *HA-Sp140* knock-in mice and depiction of resulting *HA-Sp140*[+/+] founder line. b. Immunoblot of BMMs of indicated genotypes treated with 100 µg/mL DMXAA for 8 h or 10 ng/mL IFNγ for 24 h. Actin blot represents loading control for anti-SP140 blot, and sample processing control for anti-HA blot. For gel source data, see Supplementary Fig. 1. Results representative of two independent experiments. c. RT-qPCR for *Ifnb1* from BMMs of indicated genotypes after 8 h of 100 µg/mL DMXAA treatment. Data mean +/− s.e.m are plotted, n = 3 wells of cells, * = $p < 0.05$, ** = $p < 0.005$, *** = $p < 0.0005$, **** = $p < 0.0001$, ns = not significant, one-way ANOVA with Dunnett's T3 post-hoc correction. Adjusted $p < 0.0001$ for B6 vs. *Sp140*[−/−] and *Sp140*[−/−] vs. *HA-Sp140*[+/+], and adjusted $p = 0.3071$ for B6 vs. *HA-Sp140*[+/+]. Additional exact adjusted $p$ values and statistical test results are provided in Source Data. d. Immunofluorescence of B6 BMMs treated with 8 h of 100 µg/mL DMXAA stained with anti-HA, anti-PML, and DAPI. Staining control for Fig. 5a. e. Immunofluorescence of B6 BMMs treated with 8 h of 100 µg/mL DMXAA stained with anti-HA, anti-fibrillarin, and DAPI. Staining control for Fig. 5b.

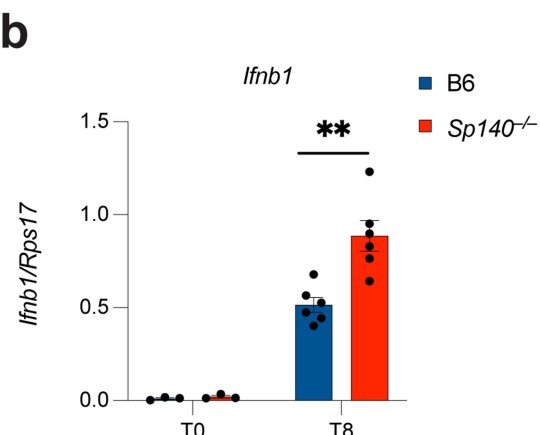

**a**

**b**

*Ifnb1*

- ■ B6
- ■ *Sp140⁻/⁻*

**Extended Data Fig. 10 | Gating strategy for BMMs infected with GFP-encoding viruses, and measurement of *Ifnb1* transcripts in B6 and *Sp140⁻/⁻* BMMs upon infection with MHV68-GFP.** a. Representative flow plots and gating strategy for BMMs infected with viruses encoding GFP. Experiment shown is for B6 BMMs infected with MHV68-GFP, MOI 3, for 24 h. b. RT-qPCR of BMMs 8 h after infection with MHV68-GFP, MOI of 1. Data mean +/− s.e.m are plotted, n = 6 wells of cells, * = $p < 0.05$, ** = $p < 0.005$, *** = $p < 0.0005$, **** = $p < 0.0001$, ns = not significant, two-tailed t-test with Welch's correction. $p = 0.333356$ for T = 0 h and 0.004352 for T = 8 h. Results representative of two independent experiments. Statistical test results are provided in Source Data.

# Reporting Summary

## Statistics

For all statistical analyses, confirm that the following items are present in the figure legend, table legend, main text, or Methods section.

| n/a | Confirmed | |
|---|---|---|
| ☐ | ☒ | The exact sample size (*n*) for each experimental group/condition, given as a discrete number and unit of measurement |
| ☐ | ☒ | A statement on whether measurements were taken from distinct samples or whether the same sample was measured repeatedly |
| ☐ | ☒ | The statistical test(s) used AND whether they are one- or two-sided *Only common tests should be described solely by name; describe more complex techniques in the Methods section.* |
| ☒ | ☐ | A description of all covariates tested |
| ☐ | ☒ | A description of any assumptions or corrections, such as tests of normality and adjustment for multiple comparisons |
| ☐ | ☒ | A full description of the statistical parameters including central tendency (e.g. means) or other basic estimates (e.g. regression coefficient) AND variation (e.g. standard deviation) or associated estimates of uncertainty (e.g. confidence intervals) |
| ☐ | ☒ | For null hypothesis testing, the test statistic (e.g. *F*, *t*, *r*) with confidence intervals, effect sizes, degrees of freedom and *P* value noted *Give P values as exact values whenever suitable.* |
| ☒ | ☐ | For Bayesian analysis, information on the choice of priors and Markov chain Monte Carlo settings |
| ☒ | ☐ | For hierarchical and complex designs, identification of the appropriate level for tests and full reporting of outcomes |
| ☒ | ☐ | Estimates of effect sizes (e.g. Cohen's *d*, Pearson's *r*), indicating how they were calculated |

*Our web collection on statistics for biologists contains articles on many of the points above.*

## Software and code

Policy information about availability of computer code

| | |
|---|---|
| Data collection | AlphaFold v2.3.2 was used to generate AlphaFold structure predictions. |
| Data analysis | Imaris File Converter v10.0.1, Imaris Stitcher v9.9.1, GraphPad Prism v10.0.2, Imaris v10.1, FlowJo 10.10.0, BWA-MEM v0.7.15, BBDuk v38.05, hisat2 v2.1.0, deepTools bamCoverage v3.0.1, Salmon v0.13.1, DESeq2 v1.38.3, ggplot2 v3.5.0, MACS2 v2.1.1, bedtools intersect v2.28.0, bedGraphToBigWig v4 ,SnapGene v7.0.1. and PAEViewer v1.0.0, Fiji v2.14.0/1.54f, Synthego ICE (https://ice.editco.bio/#/, v3.0), cistrome toolkit data browser (http://dbtoolkit.cistrome.org/), closestBed (v2.28.0), and the GREAT package v4.0.4. The mm10 genome was downloaded from https://genome.ucsc.edu/cgi-bin/hgGateway?db=mm10.  Code used in this publication is available at https://github.com/adziulko/The-SP140-RESIST-pathway. |

For manuscripts utilizing custom algorithms or software that are central to the research but not yet described in published literature, software must be made available to editors and reviewers. We strongly encourage code deposition in a community repository (e.g. GitHub). See the Nature Portfolio guidelines for submitting code & software for further information.

## Data

Policy information about availability of data

All manuscripts must include a data availability statement. This statement should provide the following information, where applicable:

- Accession codes, unique identifiers, or web links for publicly available datasets
- A description of any restrictions on data availability
- For clinical datasets or third party data, please ensure that the statement adheres to our policy

HA-SP140 anti-HA CUT&RUN data is available at GEO accession: GSE269315. RNA-seq for for Ifnar−/− and Sp140−/−Ifnar−/− BMMs is available at GEO accession: GSE269761. RNA-seq and ATAC-seq for B6 and Sp140−/− BMMs is available at GEO accession GSE269808 and GSE269811 respectively.

## Research involving human participants, their data, or biological material

Policy information about studies with human participants or human data. See also policy information about sex, gender (identity/presentation), and sexual orientation and race, ethnicity and racism.

| | |
|---|---|
| Reporting on sex and gender | This information has not been collected. |
| Reporting on race, ethnicity, or other socially relevant groupings | This information has not been collected. |
| Population characteristics | This information has not been collected. |
| Recruitment | This information has not been collected. |
| Ethics oversight | This information has not been collected. |

Note that full information on the approval of the study protocol must also be provided in the manuscript.

# Field-specific reporting

Please select the one below that is the best fit for your research. If you are not sure, read the appropriate sections before making your selection.

☒ Life sciences  ☐ Behavioural & social sciences  ☐ Ecological, evolutionary & environmental sciences

For a reference copy of the document with all sections, see nature.com/documents/nr-reporting-summary-flat.pdf

# Life sciences study design

All studies must disclose on these points even when the disclosure is negative.

| | |
|---|---|
| Sample size | We did not pre-calculate sample sizes for our experiments. We generally chose sample sizes of at least 3 for in vitro experiments based on previous work (Ji and Witt et al, 2021) demonstrating the noise and variability in vitro macrophage experiments; these sample sizes, based on p values, were sufficient to show significant differences between groups. For in vivo experiments, we used sample sizes of 4-8 mice per group per independent experiment, based on previous observations of variability in groups for Legionella pneumophila in vivo infections (Ji and Witt et al, 2021). |
| Data exclusions | For RT-qPCR, data were excluded for predetermined criteria (bad ROX reference dye annotation or low housekeeping gene amounts indicative of RNA degradation). |
| Replication | All results were reproduced in at least 2 independent experiments. |
| Randomization | Samples were assigned based on mouse genotype, or randomized for assignment to treatment groups when possible. Covariates like sex and age were controlled for in vivo experiments and bone marrow-derived macrophage generation by matching these variables across genotype groups. Experiments were designed to equalize treatment conditions for all samples. |
| Blinding | Investigators were not blinded, as all results and analyses are based on objective data that was quantified by automated instruments or image analysis software. |

# Reporting for specific materials, systems and methods

We require information from authors about some types of materials, experimental systems and methods used in many studies. Here, indicate whether each material, system or method listed is relevant to your study. If you are not sure if a list item applies to your research, read the appropriate section before selecting a response.

## Materials & experimental systems

| n/a | Involved in the study |
|-----|----------------------|
| ☐ | ☒ Antibodies |
| ☐ | ☒ Eukaryotic cell lines |
| ☒ | ☐ Palaeontology and archaeology |
| ☐ | ☒ Animals and other organisms |
| ☒ | ☐ Clinical data |
| ☒ | ☐ Dual use research of concern |
| ☒ | ☐ Plants |

## Methods

| n/a | Involved in the study |
|-----|----------------------|
| ☒ | ☐ ChIP-seq |
| ☐ | ☒ Flow cytometry |
| ☒ | ☐ MRI-based neuroimaging |

## Antibodies

| | |
|---|---|
| Antibodies used | rabbit anti-HA monoclonal antibody (Cell Signaling Technologies, C29F4; 0.5 microgram per reaction), rabbit isotype control IgG (Epicypher, 13-0042; 0.5 microgram per reaction), rat anti-HA (Roche, clone 3F10, 1186742300; 1:1000 for WB, 1:200 for IF), mouse anti-actin (Santa Cruz Biotechnology, sc-47778; 1:1000), rabbit anti-CNOT1 (Cell Signaling Technologies, 44613S; 1:1000), rabbit anti-CNOT9 (Proteintech, 22503-1-AP; 1:500), rabbit anti-TTP (Millipore Sigma, ABE285; 1:1000), rabbit anti-CNOT11 (Sigma Aldrich, HPA069823; 0.4 microgram/mL), rabbit anti-ZFP36L1 (Cell Signaling Technologies, 30894S; 1:1000), rabbit anti-ZFP36L2 (Abcam, ab70775; 1:1000), rabbit anti FLAG (Thermo Fisher Scientific, PA1-984B; 1:1000) and rabbit-SP140 (Covance, as previously described in Ji and Witt et al 2021; 1:1000). Antibodies used in IF included the following: mouse anti-PML Millipore Sigma, 05-718, 1:100; rat anti-HA, Roche, 11867423001, 1:200; rabbit anti-Fibrillarin, Abcam, ab166630, 1:100. Secondary antibodies used in the study were donkey anti-rat Alexa Fluor 488, Invitrogen, A21208; goat anti-mouse Alexa Fluor 647, Invitrogen, A21236; goat anti rabbit 647, Life technologies, A21244. All secondary antibodies were used at a 1:1000 dilution. |
| Validation | Antibodies were validated by manufacturers based on statements on the manufacturer's websites (anti-actin: https://www.scbt.com/p/beta-actin-antibody-c4; anti-PML: https://www.sigmaaldrich.com/US/en/product/mm/05718; anti-fibrillarin: https://www.abcam.com/en-us/products/primary-antibodies/fibrillarin-antibody-epr10823b-nucleolar-marker-ab166630?srsltid=AfmBOorQJAsQoGYZO8Whljr5ZQqUAFIjUy0ol8GN8bQFVCnb_zZDCHPz) and many antibodies within our publication were validated by inclusion of appropriate negative controls (for example, untagged samples were assessed for antibodies against epitope tags like HA. Anti-HA (Roche) was validated by the data included in Extended Data Figure 9b, d, and e, and Figure 5a; Anti-HA (CST) for CUT&RUN was validated by the inclusion of non HA-tagged SP140 controls (Extended Data Figure 1b). Anti-CNOT1, anti-CNOT11, and anti-CNOT9 were validated by data in Extended Data Figure 7a. Anti-TTP, anti-ZFP36L1 and anti-ZFP36L2 were validated by the data presented in Extended Data Figure 7a. Anti-FLAG (PA1-984B) was validated by the data present in Figure 3g. Anti-SP140 was validated by the data in Extended Data Figure 9b. |

## Eukaryotic cell lines

Policy information about cell lines and Sex and Gender in Research

| | |
|---|---|
| Cell line source(s) | Cell lines (HEK293Ts and GP2 cells) were obtained from the UC Berkeley TC facility. BlaER1 cells originated from the lab of Veit Hornung. Bone-marrow derived macrophages from both male and female mice were used with no impact on obtained results. |
| Authentication | All cell lines were authenticated by STR profiling. |
| Mycoplasma contamination | Cell lines were tested by PCR for mycoplasma contamination, and were not positive. |
| Commonly misidentified lines (See ICLAC register) | No commonly misidentified lines were used in this study. |

## Animals and other research organisms

Policy information about studies involving animals; ARRIVE guidelines recommended for reporting animal research, and Sex and Gender in Research

| | |
|---|---|
| Laboratory animals | This study involved the laboratory animal Mus musculus, strain C57BL6/J, and strain B6.129S2-Ifnar1tm1Agt/Mmjax, and Sp140–/– mice generated in Ji and Witt et al, Elife, 2021 on the C57BL6/J background. Sp140–/–Resist1/2–/– mice were also generated on the C57BL6/J strain background for this publication as described in publication methods. Mice were used for derivation of bone-marrow derived macrophages between 6-26 weeks of age, and for infection with Legionella pneumophila between 8 and 30 weeks of age. |
| Wild animals | The study did not involve wild animals. |
| Reporting on sex | Male and female mice were used for bone-marrow derived macrophage differentiation and for Legionella pneumophila infections, with no observable difference in obtained results based on sex. Within bone marrow macrophage derivation experiments, mice of one sex were matched across genotypes. |
| Field-collected samples | The study did not involve collection of field-samples. |

| Ethics oversight | All experiments were performed following the University of California Berkeley Institutional Animal Care and Use Committee regulatory standards. |

Note that full information on the approval of the study protocol must also be provided in the manuscript.

# Plants

| Seed stocks | Not applicable |
| Novel plant genotypes | Not applicable |
| Authentication | Not applicable |

# Flow Cytometry

## Plots

Confirm that:

☒ The axis labels state the marker and fluorochrome used (e.g. CD4-FITC).

☒ The axis scales are clearly visible. Include numbers along axes only for bottom left plot of group (a 'group' is an analysis of identical markers).

☒ All plots are contour plots with outliers or pseudocolor plots.

☒ A numerical value for number of cells or percentage (with statistics) is provided.

## Methodology

| Sample preparation | BMMs infected with MHV68-GFP were harvested in PBS, stained with Ghost Dye Far Red 780, fixed with the BD Cytofix Cytoperm kit according to kit instructions, then washed and analyzed by flow cytometry. |
| Instrument | BD LSR Fortessa X-20 and BD LSR Fortessa were used to generate flow cytometry data in this study. |
| Software | FlowJo version 10 (BD Biosciences) |
| Cell population abundance | No samples in this study were sorted. |
| Gating strategy | Samples were gated on SSC and FSC, then gated on single cells based on FSC-W and FSC-A, then gated on live cells based on Far-Red 780 staining. MHV-68 GFP positive gates were drawn based on uninfected controls. |

☒ Tick this box to confirm that a figure exemplifying the gating strategy is provided in the Supplementary Information.

