## [Peer Review File · Nature]

The SP140-RESIST pathway regulates interferon mRNA stability and antiviral immunity

Corresponding Author: Dr Russell Vance

Version 0:

Reviewer comments:

Referee #1

(Remarks to the Author)

Witt et al. describe a new IFN pathway regulator that confers post-transcriptional stability of IFN-beta. They identify two genes that are induced by SP140, which they name RESIST, based on the function of stabilizing IFN mRNA. They find that RESIST interacts with the CCR4-NOT complex to block TTP interaction. Based on previous literature and their data, they show that TTP KD stabilizes the IFNB transcript. This interesting study has implications for immunity to viruses, bacterial pathogens, and autoimmunity. The action of RESIST on the CCR4-NOT complex is novel and exciting. However, additional experimentation could strengthen this manuscript. This study will have a broad impact on the field.

1. The role of the CCR4-NOT complex in the entire transcriptome is well-established. However, it remains unclear why the RESIST activity is critical only for IFNB. This aspect of the study requires further experimental elucidation.
2. While TTP is known to bind to the AU-rich elements, biochemical data is sparse and controversial, suggesting that TTP could modulate IFNB mRNA. It would be essential to strengthen the link between TTP and IFNB by conducting biochemical experiments like RIPs or iCLIP to confirm direct interactions.
3. Does RESIST modulate other AU-rich genes such as IFN-gamma, TNF, or IL-6?
4. All the figures present mRNA levels of IFNB. However, IFN levels should also be measured by ELISA, STAT activation, and downstream ISG activation as well.
5. If the changes in the IFN expression changes are significant, they should also affect IFN-sensitive RNA virus replication.

Referee #2

(Remarks to the Author)

This study from Witt and colleagues is a fantastic study discovering a new mechanism of IFN β mRNA regulation and antiviral property of SP140. The authors identified RESIST as a new regulator of IFN β mRNA stability. However, the novelty of this study lies in the molecular mechanism of this regulation that the authors have uncovered. According to the proposed model SP140 acts as a repressor of RESIST. Upon removal of SP140-mediated repression, RESIST binds and interferes with the RNA degrading function of TTP family proteins ZFP36L1/L2. This results in stabilization of IFN β mRNA at a later time point of IRF3-mediated IFN β induction. The authors also include characterization of the antiviral property of SP140. Although the molecular mechanism of this antiviral activity is unclear. Technically, this study is very well controlled and executed with great care. Almost all the conclusions are well supported by sufficient evidence. Therefore, I think this manuscript is an appropriate candidate for publication. I have the following issues that I'd like the authors to address: The authors have only examined the antiviral activity of SP140 against MHV68, where the antiviral activity is not dependent on IFN β . However, according to the model, loss of SP140 leads to IFN β upregulation, which should promote antiviral activity against IFN-sensitive RNA viruses, such as EMCV, VSV or WNV. It will be interesting to see such results included here. Besides IFN β , several other type I IFN mRNAs also have UTR regions that results in their destabilization. In the absence of upregulation of any other IFN mRNA in SP140-null cells, the mechanism of specificity to IFN β becomes very important. It will be nice to experimentally address this issue of specificity, possibly by mapping the sequence needed for this protection mechanism to work might shed some light on its uniqueness.

Referee #3

(Remarks to the Author)

Previous work by the Vance laboratory has shown that SP140 suppresses IFN-I production through an unknown mechanism. In the new study, Witt et al found that SP140 does not inhibit the transcription of IFN β gene, but instead regulates the stability of IFN β RNA. Through gene expression analysis and other approaches, they found that SP140 represses the expression of a gene that they now name RESIST (previously named Annexin-2 receptor or ANXA2R). A previous study showed that ANXA2R interacts with CNOT1, a core component of the CCR4-NOT deadenylase complex, which functions together with the TTP family of RNA binding proteins to degrade mRNA (PMID: 30833792). Overexpression of ANXA2R was found to inhibit the replication of an RNA virus (VSV). Witt et al confirmed the interaction of RESIST with CNOT1. Importantly, they found that depletion of RESIST by sgRNA abrogated the increase of IFN β RNA in SP140 $^{-/-}$ mouse macrophages in response to stimulation with a STING agonist. Further, overexpression of SP140 enhanced the levels of IFN β RNA in both WT and SP140 $^{-/-}$ cells. The authors proposed that RESIST competes with TTP for binding with the CNOT complex, thereby inhibiting the degradation of IFN β RNA. This model is supported by their AlphaFold analysis and by their data showing that depletion of CNOT1 or TPP proteins elevated the levels of IFN β RNA even in WT cells. In addition, the authors showed that SP140 is localized in the nucleolus and exert antiviral activity. They proposed this antiviral effect of SP140 is independent of its role in regulating IFN-I.

Overall, the authors provided strong evidence that RESIST/ANXA2R is largely responsible for the increase of IFN β RNA in SP140 deficient mouse macrophages, and that this is achieved through inhibiting the degradation of IFN β RNA by the CNOT-TTP RNA degradation machinery. The paper can be improved by addressing the following questions:

- 1) Is there direct evidence that RESIST competes with the binding of CNOTs with a TTP protein?
- 2) Since the CNOT-TTP machinery degrades many RNA, why does the SP140-RESIST axis only regulate IFN β RNA stability?
- 3) Does depletion of SP140 in human cells lead to an increase of RESIST protein and IFN β RNA (i.e, is this pathway conserved in human)? What are the expression profiles of RESIST in different types of cells? From the Hubel et al paper (2019), it appears that RESIST is an ISG. Does RESIST play a role in interferon regulation in WT cells (not just SP140 $^{-/-}$ cells)? Although generating RESIST deficient mice might be beyond the scope of this paper, the authors might want to discuss the potential physiological function of RESIST (not in the absence of SP140).
- 4) Does the infection with MHV68 or other viruses in SP140 $^{-/-}$ cells lead to enhanced production of IFN β ? If so, why didn't IFN inhibit the viral replication? If not, is this unique to MHV68 or does it also apply to other viruses? Wouldn't this contradict with the authors' model that the de-repression of RESIST in SP140 $^{-/-}$ cells provide a "back-up" mechanism for defense against virus infection?
- 5) In Figure 5c, it appears that there was more MHV68-GFP in Sp140 $^{-/-}$ Ifnar $^{-/-}$ cells than in Sp140 $^{-/-}$ cells (25% vs 15%), suggesting that the IFN pathway did play an antiviral role. It would be interesting to test whether Sp140 $^{-/-}$ Resist $^{-/-}$ cells are more susceptible to virus infection than Sp140 $^{-/-}$ cells; if so, the data would be supportive of the authors' model shown in Figure 5e.
- 6) Is the antiviral role of SP140 restricted to MHV68? The impact of this study can be enhanced if the authors can provide some insights into the extent and mechanism of the antiviral role of SP140.

Version 1:

Reviewer comments:

Referee #1

(Remarks to the Author)

The authors have adequately addressed all my previous comments.

Referee #2

(Remarks to the Author)

I feel the authors have sufficiently addressed the concerns of the reviewers and the manuscript is acceptable in the current form.

Referee #3

(Remarks to the Author)

This revision has addressed most of my concerns. It appears that the function of RESIST is revealed only in the absence of SP140. Nevertheless, the authors are to be commended for doing a very nice piece of detective work in solving the mystery of how SP140 suppresses IFN production. I am supportive of publishing this paper in Nature.

We thank all the reviewers for their positive and constructive reviews. Our responses are below in italics. We have largely endeavored to respond to reviewer comments with new data. Of particular note, we now report the generation and characterization of Resist1/2 knockout mice. We also examine the role of SP140 and RESIST in additional viral infections (MCMV and Sendai Virus, in addition to MHV68 in the initial submission). Using purified recombinant proteins, we also now show a direct interaction between TTP and the IFNB1 ARE, as well as a direct interaction between RESIST and CNOT9. Numerous other figure panels were also added to address additional reviewer concerns. We hope that the reviewers are satisfied with the revised manuscript.

Referee #1 (Remarks to the Author):

Witt et al. describe a new IFN pathway regulator that confers post-transcriptional stability of IFN-beta. They identify two genes that are induced by SP140, which they name RESIST, based on the function of stabilizing IFN mRNA. They find that RESIST interacts with the CCR4-NOT complex to block TTP interaction. Based on previous literature and their data, they show that TTP KD stabilizes the IFNB transcript. This interesting study has implications for immunity to viruses, bacterial pathogens, and autoimmunity. The action of RESIST on the CCR4-NOT complex is novel and exciting. However, additional experimentation could strengthen this manuscript. This study will have a broad impact on the field.

We thank the reviewer for their enthusiastic comments and appreciate their constructive suggestions for experiments to strengthen the manuscript that we address point-by-point below.

1. The role of the CCR4-NOT complex in the entire transcriptome is well-established. However, it remains unclear why the RESIST activity is critical only for IFNB. This aspect of the study requires further experimental elucidation.

The reviewer astutely notes that very few mRNAs appear to be directly regulated by SP140-RESIST, as seen in the small number of differentially expressed mRNAs in the Sp140^{-/-}Ifnar^{-/-} vs Sp140^{-/-} RNA-seq experiment (Fig. 2a). We agree with the reviewer that CCR4-NOT regulates many more mRNAs than we observe changing in Sp140-deficient cells. Importantly, however, we do not think that RESIST is a general CCR4-NOT inhibitor; instead our data (including some new data added to the revision) suggest that RESIST acts at least in part by inhibiting TTP recruitment to CCR-NOT. Therefore, we would only expect that RESIST would affect TTP-regulated mRNAs; however, some well-known TTP-regulated transcripts such as Il6 and Tnf were also not dramatically changed in our dataset.

We hypothesized that perhaps our RNA-seq experiment had missed a relevant timepoint to see these effects. Indeed, we found that TTP and the TTP family do not profoundly regulate *Tnf* and *Il6* transcripts with 8 hours of DMXAA stimulation (Figure for reviewers, below), while *Ifnb1* transcript levels are substantially affected by disruption of TTP and the TTP family (Fig. 4f).

Therefore, we used RT-qPCR to examine the levels of TNF and IL-6 at multiple timepoints (T0, 1, 2, 4, 6, 8 10 hours post-stimulation) after DMXAA stimulation. Surprisingly, this analysis did not reveal differences in *Tnf/Il6* mRNA levels between WT and *Sp140*^{-/-} BMMs (below).

We also used RT-qPCR to examine the levels of these transcripts upon stimulation of BMMs with LPS, a strong inducer of *Tnf* and *Il6*, at multiple timepoints (T0, 1, 2, 4, 6, 8 10 hours post-stimulation). However, we still did not see differences in the levels of these transcripts (below).

Therefore, our data suggest a surprising degree of specificity of RESIST for *Ifnb1* in macrophages that we currently cannot easily explain. It seems likely that the mechanism may be quite complex. For example, it is possible that *Ifnb1* transcript co-localizes with RESIST in the cell, whereas *Tnf* and *Il-6* transcripts do not. In this scenario, RESIST would specifically inhibit recruitment of *Ifnb1* mRNAs to CCR4-NOT by TTP, whereas other transcripts would be less affected. It is also possible that the TTP family members, which we find profoundly repress *Ifnb1* transcript stability (Fig. 4f), are more specific for *Ifnb1* transcripts than *Il6* and *Tnf*. Finally, it is possible that *Ifnb1* transcript is more affected by partial inhibition of TTP function mediated by RESIST than *Tnf* and *Il6*. We feel that fully understanding the mechanistic basis of SP140-RESIST specificity is an interesting question, but one that will require significant additional biochemical and structural studies that describe the mechanism by which RESIST affects CCR4-NOT and recruitment of *Ifnb1* mRNA. It is also possible that Resist is not entirely specific for the *Ifnb1* transcript, potentially under different stimuli conditions or in other cell types. We have included additional text in the discussion addressing the specificity of RESIST.

2. While TTP is known to bind to the AU-rich elements, biochemical data is sparse and controversial, suggesting that TTP could modulate IFNB mRNA. It would be essential to strengthen the link between TTP and IFNB by conducting biochemical experiments like RIPs or iCLIP to confirm direct interactions.

We thank the reviewer for this constructive suggestion. In our manuscript we refer to a previously published PAR-CLIP study that characterized TTP-bound transcripts (Sedlyarov, 2016). The data reported in this manuscript show unmistakable binding of TTP to the *Ifnb1* 3' UTR (see below).

This peak coincides with the AU-rich element (ARE) present in the 3' UTR of the Ifnb1 transcript, which is a well-characterized TTP target sequence. In addition, in line with our results, Sedlyarov et al also observe increased Ifnb1 transcript levels in TTP KO cells with LPS stimulation. Additionally, using recombinant purified TTP in an in vitro assay, in new data added to the revision, we found that the TTP RNA-binding zinc finger domain clearly binds the Ifnb1 3'UTR in a manner dependent upon the ARE (Extended Data 8). Taken together, we feel these data provide strong evidence for direct regulation of IFNB1 by TTP.

3. Does RESIST modulate other AU-rich genes such as IFN-gamma, TNF, or IL-6?

This is a great question which essentially restates the same question as reviewer comment #1, and is addressed in more detail above for TNF and IL-6. In brief, we examined TNF and IL-6 mRNA levels as suggested by the reviewer and did not observe significant modulation by SP140-RESIST. As acknowledged above, this does seem to indicate a surprising degree of specificity, which we feel will require substantial additional mechanistic experimentation to fully understand. Since our study focused on macrophages, which do not produce IFN-gamma, we did not examine that transcript.

4. All the figures present mRNA levels of IFNB. However, IFN levels should also be measured by ELISA, STAT activation, and downstream ISG activation as well.

Multiple papers from our group have previously shown that SP140 deficient mice exhibit elevated IFN-I levels and ISG activation (in vitro and in vivo) and confirm that this interferon is biologically meaningful in driving susceptibility of these mice to bacterial infections (Ji and Witt et al, 2021; Ji et al, 2019). To address the reviewer's concern, we have added new data showing an increase in IFN β protein produced by Sp140^{-/-} BMMs in Figure 1d. To show that RESIST also regulates IFN β protein levels, we have added new data showing that Sp140^{-/-} Resist1^{-/-} Resist2^{-/-} BMMs secrete similar levels of IFN β protein as WT BMMs (Figure 3d). We also find that Sp140^{-/-} BMMs restrict MCMV

and Sendai virus replication (Figure 5f,g). This restriction depends on RESIST, which suggests the increased IFN β protein secreted by Sp140^{-/-} BMMs is biologically meaningful in countering viral replication. Finally, we show that RESIST drives the susceptibility of Sp140^{-/-} mice to Legionella infection in vivo (Figure 3i), which we have previously reported to be driven by IFN-I (Ji and Witt et al 2021).

5. If the changes in the IFN expression changes are significant, they should also affect IFN-sensitive RNA viruses replication.

We thank the reviewer for this suggestion, which turned out to be a great one. As a new experiment added to the revision, we infected macrophages with Sendai-GFP (an RNA virus encoding GFP) and found a defect in viral replication in Sp140^{-/-} BMMs as expected, which depends on RESIST (Figure 5g).

Referee #2 (Remarks to the Author):

This study from Witt and colleagues is a fantastic study discovering a new mechanism of IFN β mRNA regulation and antiviral property of SP140. The authors identified RESIST as a new regulator of IFN β mRNA stability. However, the novelty of this study lies in the molecular mechanism of this regulation that the authors have uncovered. According to the proposed model SP140 acts as a repressor of RESIST. Upon removal of SP140-mediated repression, RESIST binds and interferes with the RNA degrading function of TTP family proteins ZFP36L1/L2. This results in stabilization of IFN β mRNA at a later time point of IRF3-mediated IFN β induction. The authors also include characterization of the antiviral property of SP140. Although the molecular mechanism of this antiviral activity is unclear. Technically, this study is very well controlled and executed with great care. Almost all the conclusions are well supported by sufficient evidence. Therefore, I think this manuscript is an appropriate candidate for publication. I have the following issues that I'd like the authors to address:

We thank the reviewer for their generous comments. We address their specific remaining concerns below.

1. The authors have only examined the antiviral activity of SP140 against MHV68, where the antiviral activity is not dependent on IFN β . However, according to the model, loss of SP140 leads to IFN β upregulation, which should promote antiviral activity against IFN-sensitive RNA viruses, such as EMCV, VSV or WNV. It will be interesting to see such results included here.

We thank the reviewer for this suggestion. Reviewer 1 (comment 5) made a similar suggestion. In new data added to the revision, we found that Sp140^{-/-} BMMs exhibit enhanced resistance to MCMV-GFP and Sendai-GFP virus replication, which in both instances depends on RESIST (Figure 5f, g). We also show that IFNAR is required for the resistance of Sp140^{-/-} cells to MCMV, and in addition, that IFNAR enhances the resistance of Sp140^{-/-} cells to MHV-68. Together these results provide strong support

for the reviewer's hypothesis that loss of SP140 can enhance viral resistance via increased levels of IFN β .

2. Besides IFN β , several other type I IFN mRNAs also have UTR regions that results in their destabilization. In the absence of upregulation of any other IFN mRNA in SP140-null cells, the mechanism of specificity to IFN β becomes very important. It will be nice to experimentally address this issue of specificity, possibly by mapping the sequence needed for this protection mechanism to work might shed some light on its uniqueness.

It is an interesting idea that perhaps SP140-RESIST might regulate other type I IFNs. In our RNA-seq of DMXAA-treated Sp140^{-/-} BMMs, Ifna2, Ifna4, and Ifna5 are upregulated with very minor significance. We compared the mRNAs of these interferons to that of Ifnb1. However, all of these mRNAs are quite similar in sequence and we did not observe obvious elements unique to Ifnb1 or the other IFNs that could explain the specificity of RESIST for Ifnb1. We are therefore unable to determine whether the apparent specificity of RESIST for Ifnb1 derives from the subtle differences in sequence between the mRNA transcripts of Ifnb1 and the other IFNs or simply because other IFNs besides Ifnb1 are not robustly induced by DMXAA in BMMs. To respond to the reviewer's comment about mapping the relevant element, and as described above in response to reviewer 1 (comment 2), we did identify an ARE in the 3'UTR of the IFN β mRNA and experimentally demonstrate that this element binds specifically to TTP as expected (Extended data Fig 8). This ARE is not dramatically different in sequence from that in the 3'UTR of Ifna mRNAs; however, it is possible there are functional differences, especially with respect to whether these Ifna mRNA isoforms bind to TTP homologs or perhaps to other regulators of mRNA stability (e.g., Roquin, which is not antagonized by RESIST). In sum, the regulation of interferon mRNA stability is likely to be highly complex and will likely require an in depth analysis in separate work.

Referee #3 (Remarks to the Author):

Previous work by the Vance laboratory has shown that SP140 suppresses IFN-I production through an unknown mechanism. In the new study, Witt et al found that SP140 does not inhibit the transcription of IFN β gene, but instead regulates the stability of IFN β RNA. Through gene expression analysis and other approaches, they found that SP140 represses the expression of a gene that they now name RESIST (previously named Annexin-2 receptor or ANXA2R). A previous study showed that ANXA2R interacts with CNOT1, a core component of the CCR4-NOT deadenylase complex, which functions together with the TTP family of RNA binding proteins to degrade mRNA (PMID: 30833792). Overexpression of ANXA2R was found to inhibit the replication of an RNA virus (VSV). Witt et al confirmed the interaction of RESIST with CNOT1. Importantly, they found that depletion of RESIST by sgRNA abrogated the increase of IFN β RNA in SP140^{-/-} mouse macrophages in response to stimulation with a STING agonist. Further, overexpression of SP140 enhanced the levels of IFN β RNA in both WT and SP140^{-/-} cells. The authors proposed that RESIST competes with TTP for binding with the CNOT complex, thereby inhibiting the degradation of IFN β RNA. This model is

supported by their AlphaFold analysis and by their data showing that depletion of CNOT1 or TPP proteins elevated the levels of IFN β RNA even in WT cells. In addition, the authors showed that SP140 is localized in the nucleolus and exert antiviral activity. They proposed this antiviral effect of SP140 is independent of its role in regulating IFN-I.

Overall, the authors provided strong evidence that RESIST/ANXA2R is largely responsible for the increase of IFN β RNA in SP140 deficient mouse macrophages, and that this is achieved through inhibiting the degradation of IFN β RNA by the CNOT-TTP RNA degradation machinery. The paper can be improved by addressing the following questions:

1) Is there direct evidence that RESIST competes with the binding of CNOTs with a TTP protein?

We thank the reviewer for their positive assessment and for their constructive suggestions. We also think showing evidence of competition between RESIST and TTP for CCR4-NOT binding is important for our model. We have therefore included new data showing that co-expression of RESIST with TTP inhibits the interaction of TTP with the CCR4-NOT complex (Figure 4g). We have also included a pulldown with purified recombinant proteins that shows that RESIST directly interacts with the CNOT9 subunit of CCR4-NOT (Figure 4c). This result is particularly important as it is supported by genetic data showing an essential role for CNOT9 in the activity of RESIST, and in addition, supports a direct mechanism for RESIST in modulation of CCR4-NOT.

2) Since the CNOT-TTP machinery degrades many RNA, why does the SP140-RESIST axis only regulate IFN β RNA stability?

*This is an excellent point that was raised by all three reviewers. We direct the reviewer to our detailed response to Reviewer 1 (comment 1). In brief, our followup work confirms a surprising degree of specificity for RESIST in regulation of *Ifnb1*. However, the underlying mechanism appears likely to be complex. We have discussed various possibilities in the revised text, but to properly investigate these possibilities would require considerable additional experimentation that we feel is best left to a separate publication.*

3) (a) Does depletion of SP140 in human cells lead to an increase of RESIST protein and IFN β RNA (i.e, is this pathway conserved in human)? (b) What are the expression profiles of RESIST in different types of cells? (c) From the Hubel et al paper (2019), it appears that RESIST is an ISG. Does RESIST play a role in interferon regulation in WT cells (not just SP140-/- cells)? (d) Although generating RESIST deficient mice might be beyond the scope of this paper, the authors might want to discuss the potential physiological function of RESIST (not in the absence of SP140).

*(a) To address the reviewers question, we tested SP140 KO in human immortalized *Bl*a*ER1* macrophages and we did not see an effect on RESIST or IFNB1 expression.*

However, we also tested deficiency of SP140 in immortalized mouse macrophages and also did not see an effect. Therefore, it may be that the SP140-RESIST-IFN pathway does not function normally in immortalized cells. To address this, we attempted to knock out SP140 in primary human monocytes. Again, we did not observe differences in IFNB1 transcript levels after 8 hours of ADU-S100 stimulation. However, our SP140 knockout efficiency was quite low (~30%) and we did not recover enough cells after electroporation to test multiple timepoints. An additional complication is that humans have an extra SP family member (SP140L) and may have more functional redundancy within this family than mice. To see an effect on IFNB1 transcript levels, it may be necessary to target all 4 SP family members in primary human cells, which is extremely difficult. We feel this experiment would require extensive optimization that will be interesting to pursue in future work. It is worth emphasizing, though, that we were able to see that RESIST overexpression positively regulates IFNB1 expression in human cells by about 10-fold (Fig 3h). Thus, we feel our findings are relevant for human cells, though we acknowledge there may be regulatory differences between mice and humans.

(b) We have now included expression profiles of RESIST in mice and humans in Extended Data Fig. 3.

(c) Based on our results in Figure 3b and 3e, we do not believe RESIST plays a large role in interferon regulation in WT macrophages, likely because it is transcriptionally repressed by SP140. We now clearly discuss this in the Discussion.

(d) In what we consider to be a substantial addition to the manuscript, we now include data from newly generated *Sp140^{-/-}Resist1/2^{-/-}* mice. These results validate that RESIST plays a critical physiological role in regulating IFN in macrophages in vitro (Fig 3c-d) and in driving susceptibility of *Sp140^{-/-}* mice to bacterial infection (Figure 3i). We have included more discussion of the potential physiological roles of the RESIST pathway in the discussion. While we don't see a role for IFN-I regulation by RESIST in macrophages, we speculate that RESIST could play an antiviral role even in the presence of SP140 by stabilizing IFN-I or other cytokine mRNAs in vivo during viral infection, as RESIST is an ISG.

4) Does the infection with MHV68 or other viruses in SP140^{-/-} cells lead to enhanced production of IFN β ? If so, why didn't IFN inhibit the viral replication? If not, is this unique to MHV68 or does it also apply to other viruses? Wouldn't this contradict with the authors' model that the de-repression of RESIST in SP140^{-/-} cells provide a "back-up" mechanism for defense against virus infection?

In a new experiment added to the revision, we measured *Ifnb1* transcript levels in WT and *Sp140^{-/-}* BMMs upon MHV68 infection and found that *Sp140^{-/-}* BMMs show enhanced *Ifnb1* induction (Extended Data Fig. 10b). We think the elevated IFN-I produced by *Sp140^{-/-}* BMMs does repress MHV68 replication, as viral replication is increased in *Sp140^{-/-} Resist1^{-/-} Resist2^{-/-}* BMMs and *Sp140^{-/-} Ifnar^{-/-}* compared to *Sp140^{-/-}* BMMs (Fig. 5d-e). We also find that elevated IFN-I mediated by RESIST

restricts replication of other viruses like MCMV and Sendai in Sp140^{-/-} BMMs (new data, Fig. 5f-g). Ultimately, we think that while RESIST induction mitigates the loss of SP140 as an important antiviral protein, ultimately the role of SP140 as an MHV68 restriction factor outweighs the increased IFN-I produced by Sp140^{-/-} BMMs in MHV68-GFP infection. For other viral infections, like MCMV and Sendai, the antiviral activity of SP140 is less important and instead the increased IFN-I present in Sp140^{-/-} BMMs restricts viral replication.

5) In Figure 5c, it appears that there was more MHV68-GFP in Sp140^{-/-}Ifnar^{-/-} cells than in Sp140^{-/-} cells (25% vs 15%), suggesting that the IFN pathway did play an antiviral role. It would be interesting to test whether Sp140^{-/-}Resist^{-/-} cells are more susceptible to virus infection than Sp140^{-/-} cells; if so, the data would be supportive of the authors' model shown in Figure 5e.

As discussed above in our response to comment #4, we tested whether Sp140^{-/-} Resist1^{-/-} Resist2^{-/-} BMMs were more susceptible to MHV68-GFP and other viruses. Indeed, we found that a lack of RESIST makes Sp140^{-/-} BMMs more susceptible to infection with multiple viruses (MHV68, MCMV, Sendai) (Figure 5e-g), which we agree is supportive of our model shown in Figure 5h.

6) Is the antiviral role of SP140 restricted to MHV68? The impact of this study can be enhanced if the authors can provide some insights into the extent and mechanism of the antiviral role of SP140.

As discussed above, we infected WT and Sp140^{-/-} BMMs with another herpesvirus (MCMV) to determine the extent of the antiviral activity of SP140 (Fig. 5f). Interestingly, we did not see an antiviral activity of SP140 against MCMV; thus, there does appear to be some selectivity of SP140 for different viruses. We currently do not understand the basis for this selectivity, but it is not unusual for an antiviral factor to have specificity for one virus versus another. As one possible mechanism to explain why SP140 does not restrict MCMV, we hypothesize SP140 may be antagonized by MCMV viral effectors that efficiently target SP140 and neutralize its antiviral activity. As we refer to in the text, there are many examples of viral effectors that antagonize nuclear body proteins. We feel that fully dissecting the mechanism by which SP140 is antiviral will be extremely involved and is beyond the scope of this paper. Multiple papers have been published on the antiviral mechanism of the SP140-like protein SP100, yet much remains to be learned about how SP100 truly functions as an antiviral protein—it is certainly not a straightforward question. We have expanded our discussion of how SP140 may be antiviral in the discussion.